# A high-throughput cytotoxicity screening platform reveals *agr*-independent mutations in bacteraemia-associated *Staphylococcus aureus* that promote intracellular persistence

Abderrahman Hachani[1]*[†], Stefano G Giulieri[1†], Romain Guérillot[1†], Calum J Walsh[1], Marion Herisse[1], Ye Mon Soe[1], Sarah L Baines[1], David R Thomas[1,2], Shane Doris Cheung[3], Ashleigh S Hayes[1], Ellie Cho[3], Hayley J Newton[1,2], Sacha Pidot[1], Ruth C Massey[4,5,6,7], Benjamin P Howden[1,8‡], Timothy P Stinear[1‡]

[1]Department of Microbiology and Immunology, Doherty Institute, University of Melbourne, Melbourne, Australia; [2]Infection and Immunity Program, Department of Microbiology and Biomedicine Discovery Institute, Monash University, Clayton, Australia; [3]Biological Optical Microscopy Platform, University of Melbourne, Melbourne, Australia; [4]School of Microbiology, University College Cork, Cork, Ireland; [5]School of Medicine, University College Cork, Cork, Ireland; [6]APC Microbiome Ireland, University College Cork, Cork, Ireland; [7]School of Cellular and Molecular Medicine, University of Bristol, Bristol, United Kingdom; [8]Microbiological Diagnostic Unit Public Health Laboratory, Department of Microbiology and Immunology, Doherty Institute, University of Melbourne, Melbourne, Australia

*For correspondence:
abderrahman.hachani@unimelb.edu.au

[†]These authors contributed equally to this work
[‡]These authors also contributed equally to this work

Competing interest: The authors declare that no competing interests exist.

**Abstract** *Staphylococcus aureus* infections are associated with high mortality rates. Often considered an extracellular pathogen, *S. aureus* can persist and replicate within host cells, evading immune responses, and causing host cell death. Classical methods for assessing *S. aureus* cytotoxicity are limited by testing culture supernatants and endpoint measurements that do not capture the phenotypic diversity of intracellular bacteria. Using a well-established epithelial cell line model, we have developed a platform called *InToxSa* (intracellular toxicity of *S. aureus*) to quantify intracellular cytotoxic *S. aureus* phenotypes. Studying a panel of 387 *S. aureus* bacteraemia isolates, and combined with comparative, statistical, and functional genomics, our platform identified mutations in *S. aureus* clinical isolates that reduced bacterial cytotoxicity and promoted intracellular persistence. In addition to numerous convergent mutations in the Agr quorum sensing system, our approach detected mutations in other loci that also impacted cytotoxicity and intracellular persistence. We discovered that clinical mutations in *ausA*, encoding the aureusimine non-ribosomal peptide synthetase, reduced *S. aureus* cytotoxicity, and increased intracellular persistence. *InToxSa* is a versatile, high-throughput cell-based phenomics platform and we showcase its utility by identifying clinically relevant *S. aureus* pathoadaptive mutations that promote intracellular residency.

## Editor's evaluation

This paper describes a new method to investigate *Staphylococcus aureus* intracellular virulence that has produced important insights into the mechanisms of staphylococcal pathogenesis. The results are convincing and the methodology is state-of-the-art. The authors have responded to the reviewer

comments and resolved the issues identified during the review. This paper will be of interest to scientists studying microbial intracellular pathogenesis and cell biology.

## Introduction

*Staphylococcus aureus* is a leading cause of hospital-acquired infections, a problem exacerbated by increasing resistance to last-line antibiotics (*Murray et al., 2022*; *Tong et al., 2015*). *S. aureus* is traditionally considered an extracellular pathogen as it produces many secreted virulence factors, including superantigens, degradative enzymes, and cytolytic toxins (*Tam and Torres, 2019*). The potent pore-forming toxins (PFTs), including alpha-hemolysin and leukocidins, are among the major virulence determinants of *S. aureus* (*Cheung et al., 2021*). These toxins induce rapid host cell death, including death of the leukocytes and neutrophils recruited to remove bacteria from infected tissues (*Surewaard et al., 2016*; *Thammavongsa et al., 2015*).

Long-term asymptomatic *S. aureus* carriage is common but invasive infection is rare. Thus, understanding the changes enabling *S. aureus* to switch from a common coloniser of the nasal cavity to an invasive pathogen is a major research focus. A powerful discovery approach has been used to compare the cytolytic activities of secreted *S. aureus* virulence factors between colonising and bacteraemia isolates, followed by genomics to uncover the bacterial genetic loci linked with high/low cytolytic activity (*Collins et al., 2008*; *Giulieri et al., 2018*; *Laabei et al., 2014*; *Laabei et al., 2021*; *McConville et al., 2022*). These toxicity analyses use methods that monitor eukaryotic cell viability upon exposure to *S. aureus* culture supernatants (*Dankoff et al., 2020*; *Das et al., 2016*; *Giulieri et al., 2018*) and integrate these phenotypes within genome-wide association studies (GWAS) and other comparative bacterial population genomics techniques (*Giulieri et al., 2018*; *Recker et al., 2017*). However, such toxicity assays are limited in that they focus on exogenous virulence factors that have accumulated in the culture media during bacterial growth. Thus, such methods can be limiting as they report phenotypes that are temporally skewed and ignore the host cell-bacterium context under which the production of these factors is controlled during infection. When these endpoint toxicity assays are conducted at scale with many hundreds of bacterial isolates, additional issues caused by filter-sterilisation, freeze-storage, and other manipulations during the preparation of bacterial supernatants prior to incubation with host cells may increase assay variability (*Giulieri et al., 2018*; *McConville et al., 2022*).

Another restriction of toxicity assays is that they treat *S. aureus* as an obligate extracellular pathogen, whilst the literature has reported its capacity to adopt an intracellular behaviour, readily adhering to and invading various eukaryotic cells, including non-professional phagocytes (*Flannagan et al., 2016*; *Foster et al., 2014*; *Soe et al., 2021*). Upon internalisation, intracellular *S. aureus* initially reside in phagolysosomes, where low pH is a cue for bacterial replication (*Flannagan et al., 2018*; *Lâm et al., 2010*). *S. aureus* uses its arsenal of PFTs to escape from this degradative compartment, into the cytosol and cause the death of host cells (*Moldovan and Fraunholz, 2019*; *Siegmund et al., 2021*). This intracellular niche confers protection to *S. aureus* from antibiotics and immune responses (*Peyrusson et al., 2020*; *Strobel et al., 2016*). While guarding *S. aureus* from host attack, intracellular residency also enables the creation of a reservoir for the pathogen to persist in an infected host and could lead to bacterial transmigration into distal host tissues, from where the bacteria can cause protracted infections and more serious disease (*Jorch et al., 2019*; *Surewaard et al., 2016*). Toxicity and persistence of *S. aureus* in an intracellular context are critical pathogenesis traits. However, understanding these traits and their microevolution across diverse collections of clinical *S. aureus* strains, and correlating them with specific bacterial genetic variations has been hampered by the lack of a high-throughput platform for trait characterisation.

To address these issues, we took advantage of the capacity of *S. aureus* to invade cultured epithelial cell lines and established a 96-well assay format to accurately monitor over time the bacterial toxicity exerted from within host cells. HeLa cells are adherent epithelial cells and represent an amenable model to study the pathogenesis of most intracellular bacteria causing human disease, including *S. aureus* (*Das et al., 2016*; *Stelzner et al., 2020a*; *Stelzner et al., 2020b*). We modified an antibiotic/enzyme protection assay, using a combination of gentamicin and lysostaphin, to kill extracellular bacteria while preserving the viability of intracellular bacteria (*Kim et al., 2019*). Using propidium iodide (PI) as a marker of host cell death, the assay measured changes in fluorescence of HeLa cells

over time caused by the presence of intracellular *S. aureus*. Combined with the power of bacterial genomics and evolutionary convergence analysis, we used the assay to screen a large collection of *S. aureus* bacteraemia isolates. Our large-scale pheno-genomics approach revealed known and previously undescribed loss-of-function mutations that were significantly associated with reduced intracellular cytotoxicity and increased intracellular persistence.

## Results

### *InToxSa* assay development

We set out to develop a high-throughput and continuous cell death assay that could measure the intracellular toxicity of *S. aureus* in a format we named *InToxSa*. We used adherent HeLa-CCL2 epithelial cells as an infection model in a 96-well format and infected them with *S. aureus* at a standardised multiplicity of infection (MOI). Following an infection period of 2 hr, infected cells were treated with an antibiotic/enzyme combination to specifically kill extracellular bacteria and prevent further reinfection by bacteria released by cells dying during the assay. HeLa cell viability was continuously monitored by measuring PI fluorescence (see methods). Reduced HeLa cell viability was indicated by increasing fluorescence over time (*Figure 1A*). We used regression to fit standardised curves to the PI uptake data and calculated seven kinetic parameters including the area under the curve (AUC) representing the total of PI uptake over time, peak PI uptake [$\mu^{max}$], the time to $\mu^{max}$ [$t(\mu^{max})$], the maximum rate in PI uptake [$r^{max}$], the time to $r^{max}$ [$t(r^{max})$], trough, and time to trough (*Figure 1—figure supplement 1*).

We then mapped out a series of experiments to validate *InToxSa* and explored bacterial genetic factors linked to intracellular cytotoxicity (*Figure 1B*). To demonstrate method performance, we measured the intracellular toxicity of the wild-type *S. aureus* USA300 strain JE2 against an isogenic *agrA* mutant (Nebraska Transposon Library mutant NE1532; *Fey et al., 2013*), using non-infected cells as a baseline (*Figure 1C*). *S. aureus* JE2 caused a rapid and substantial increase in PI fluorescence over time, reflective of the known high cytotoxicity of this strain (*Das et al., 2016*; *Grosz et al., 2014*; *McConville et al., 2022*). Cells infected with the *agrA* mutant yielded significantly lower PI uptake (AUC) and slower ($r^{max}$), indicating HeLa cell viability during the infection course and is consistent with the reported low cytotoxicity of *S. aureus agr* mutants (*Figure 1C*; *Laabei et al., 2021*; *McConville et al., 2022*).

We then assessed the reproducibility and repeatability of *InToxSa* across five experimental replicates, each time using both biologically independent HeLa cell culture passages and independent *S. aureus* cultures with the same two comparator strains (JE2 wild type and the isogenic *agrA* transposon mutant) (*Figure 1C*). We measured seven PI uptake curve kinetic parameters (*Figure 1C*, *Figure 1—figure supplement 1*). We observed that the PI uptake AUC, peak PI uptake [$\mu^{max}$], and maximum PI uptake rate [$r^{max}$] for *S. aureus* JE2 compared to the *agrA* mutant and non-infected cells were significantly different, had very low intra-assay variation, and were among the most discriminatory and reproducible PI uptake curve parameters (*Figure 1D*, *Table 1*, *Figure 1—figure supplement 1*). At experimental endpoints, the acidity of the culture media had not changed, and no bacterial growth was observed after plating the media from infected 96-well plates, indicating that *InToxSa* assessed the cytotoxicity caused only by intracellular *S. aureus* (data not shown).

### Confirmation of *S. aureus* internalisation in HeLa cells

We used confocal microscopy to confirm the presence of intracellular *S. aureus* (*Figure 2A*). HeLa cells grown on coverslips were infected with the same conditions used for *InToxSa* and analysed at 3, 8, and 24 hr post-infection. These timepoints were selected to reflect the key *InToxSa* timepoints highlighted during JE2 infection (start of PI uptake measurement, peak PI uptake [$\mu^{max}$], and experimental endpoint, respectively). We observed that at each timepoint, both JE2 and the *agrA* mutant co-localised with the lysosomal marker LAMP-1. However, the *agrA* mutant was detected in higher numbers within cells compared to wild type (*Figure 2A, B*). At later timepoints (8 and 24 hr) the number of JE2-infected cells decreased, with fewer detectable intracellular bacteria, suggesting that JE2-infected cells had died, consistent with bacterial cytotoxicity. In contrast, the number of cells infected with the *agrA* mutant did not vary significantly, indicating cell survival during the infection time course (*Figure 2B*). It also appeared that the number of intracellular bacteria increased for cells infected with the *agrA* mutant, suggesting bacterial replication over time. This latter observation was explored in

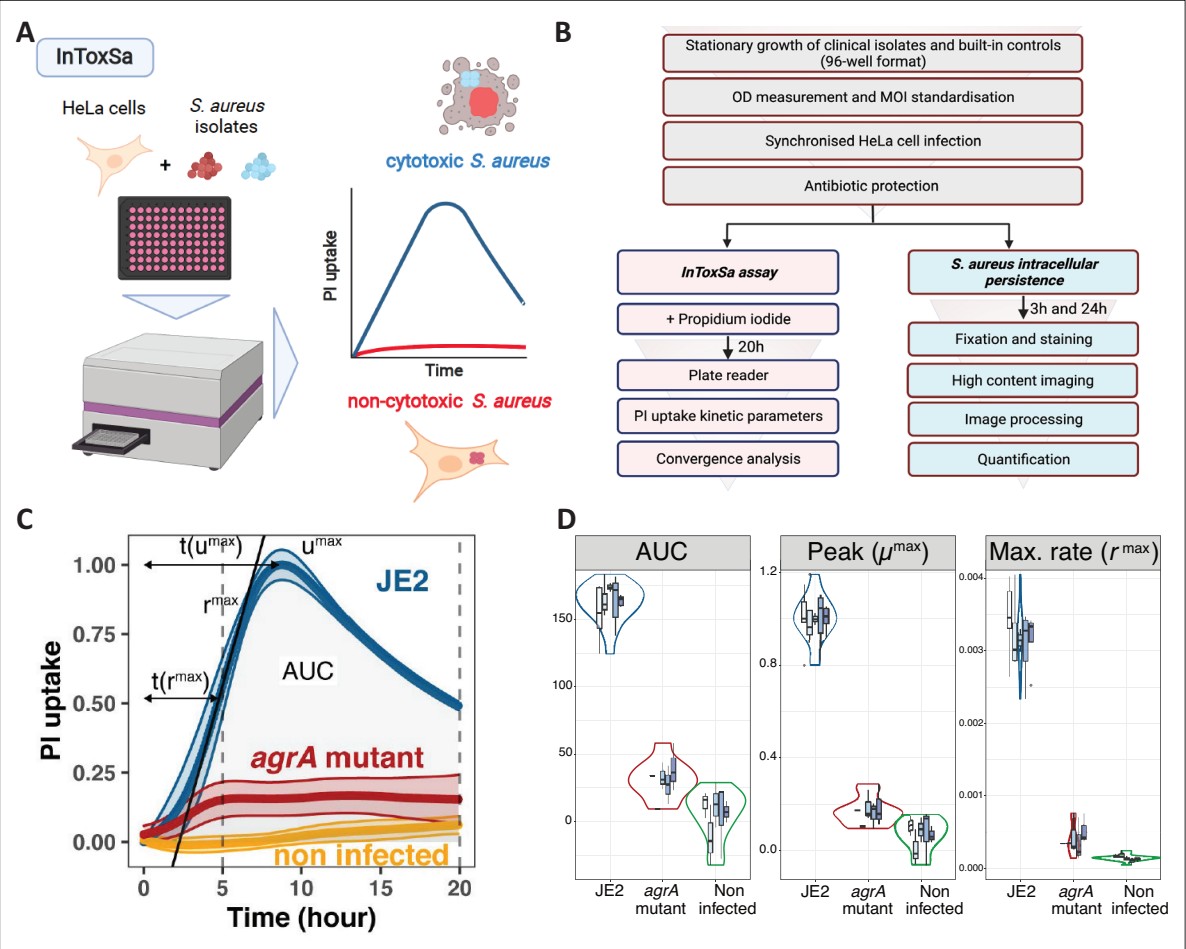

**Figure 1.** Establishing the intracellular toxicity of *S. aureus* (*InToxSa*) assay. (**A**) Overview of *InToxSa* assay. (**B**) Flow chart of the analytical pathway. (**C**) Plot of propidium iodide (PI) fluorescence uptake over 20 hr as a measure of *S. aureus* intracellular cytotoxicity in HeLa cells. Depicted curves are wild-type *S. aureus* JE2 (blue), the isogenic *S. aureus* JE2 agrA transposon mutant (red), and uninfected cells (yellow). The PI uptake curve for JE2 is annotated with five kinetic parameters. For each curve, the thick line represents the mean and the shading, the standard deviation. Curves are fitted with cubic smooth splines (see methods). To minimise batch effect, all kinetics data have been transformed using proportion of maximum scoring (POMS) using JE2 controls as reference minimum and maximum values (***Little, 2013***). *x*-axis is time and *y*-axis is PI uptake, represented as a proportion of maximal fluorescence in JE2-infected cells, where for every measured plate, a PI uptake value of 1 represents the maximum of JE2 PI uptake and zero its minimum. (**D**) Summary of five independent *InToxSa* experiments to assess assay and parameter variation. Violin plots represent the density distribution of all five replicates (based on Gaussian kernel density estimation) and the nested box plots (boxes represent median, first and third quartiles, whiskers represent are based on ×1.5 interquartile range) show the distribution of within plate replicates (3–5 technical replicates per plate replicate) for the three most discriminatory of seven parameters inferred from the PI uptake data (***Figure 1—figure supplement 1***).

The online version of this article includes the following figure supplement(s) for figure 1:

**Figure supplement 1.** Propidium iodide (PI) uptake curve parameters.

more depth using high-content/high-throughput imaging (see later section). The microscopy results support the *InToxSa* assay outputs and indicate that non-cytotoxic *S. aureus*, such as the *agrA* mutant, can persist within cells without affecting cell viability.

## *InToxSa* benchmarking against trypan blue exclusion assay

In a previous study, we used a trypan blue exclusion assay with THP1 human macrophages exposed to culture supernatants to test within-host cytotoxicity variations from 51 clinical *S. aureus* isolated from 20 patients with bacteraemia (***Giulieri et al., 2018***). These 51 *S. aureus* isolates were originally selected because they were associated with phenotypic changes occurring during bacteraemia, such as infection relapse, antibiotic treatment failure, longer duration of bacteraemia, and augmented vancomycin MIC; phenotypes likely resulting from within-host evolution (***Giulieri et al., 2018***). Thus,

**Table 1.** Summary of intracellular toxicity of S. aureus (*InToxSa*) assay performance.

| Strain | No. biological replicates | Area under the curve [AUC] | | | Peak uptake [$\mu^{max}$] | | | Max uptake [$r^{max}$] | | |
|---|---|---|---|---|---|---|---|---|---|---|
| | | Mean | Std. dev. | CoV | Mean | Std. dev. | CoV | Mean | Std. dev. | CoV |
| *S. aureus* JE2 wild type | 25 | 162 | 14.6 | 0.09 | 1 | 0.09 | 0.09 | 0.003 | 0.0004 | 0.13 |
| *S. aureus* JE2 *agrA* mutant | 11 | 30.9 | 14 | 0.45 | 0.17 | 0.06 | 0.35 | 0.0004 | 0.0002 | 0.52 |
| Non-infected | 15 | 5.3 | 18.3 | 3.46 | 0.07 | 0.07 | 1.02 | 0.0001 | 0.00005 | 0.32 |

Note: Std. dev. = standard deviation; CoV = coefficient of variation.

the isolates were categorised by patient and by their sequential isolation, with the first isolate considered as 'baseline' and the subsequent isolates designated as 'evolved'.

All 51 isolates were screened with *InToxSa* and compared with the original trypan blue assay data (*Figure 3*). Using trypan blue exclusion, only the evolved isolate from patient 50 (P_50) showed a significant phenotypic difference in cytotoxicity (*Figure 3A*). This difference was attributed to a loss-of-function mutation in *agrA* (T88M). *InToxSa* also identified a significant cytotoxicity difference for the P_50 isolate pair, validating the previous observation, despite the methodological and host cell type differences. However, *InToxSa* detected significantly reduced cytotoxicity for the evolved isolates from five more patient pairs compared to the original trypan blue screen (*Figure 3B*). These results support the higher sensitivity of *InToxSa* in uncovering *S. aureus* cytotoxicity variations resulting from the evolution of the bacterial population during bloodstream infection.

## Screening a large collection of clinical *S. aureus* to evaluate intracellular cytotoxicity

A major motivation for developing the *InToxSa* assay was to develop a pheno-genomics platform to efficiently measure the intracellular cytotoxicity profiles of a large collection of clinical *S. aureus* isolates and then use the power of comparative and statistical genomics to find bacterial genetic loci associated with intracellular cytotoxicity. To this end, we analysed 387 clinical *S. aureus* isolates, obtained from 298 episodes of bacteraemia and for which genome sequences were available (*Giulieri et al., 2018*; *Holmes et al., 2011*; *Holmes et al., 2014*; *Holmes et al., 2018*). A 164,449 single nucleotide polymorphism (SNP) core-genome phylogeny was inferred for this collection. The 387 isolates spanned 32 sequence types (STs) and were dominated by ST239, accounting for 30% of isolates (n:117), followed by ST22 (n:32, 8%), ST5 (n:34, 8%), ST45 (n:28, 7%), and ST1 (n:18, 5%). Fifty-three percent of the isolates were *mecA* positive (*Figure 4A*; *Supplementary file 3*).

We assessed each of the 387 isolates by *InToxSa*, with the cytotoxicity phenotype for each isolate represented by mean PI uptake (AUC) and displayed as a heatmap, aligned with the core-genome phylogeny (*Figure 4A*). Several patterns were readily seen in the data. There was a wide range of cytotoxicity profiles across the 387 isolates, with notable variations within each ST suggesting frequent adaptation events affecting intracellular cytotoxicity levels. Two STs (ST239 and ST22) strongly associated with lower cytotoxicity. These variations are consistent with previous observations (*Laabei et al., 2021*).

To select the most suitable *InToxSa* parameters for our statistical genomics approach, we assessed their relative importance by fitting the data using unsupervised random forest (RF) machine learning (*Mantero and Ishwaran, 2021*). In this model, 'observations' were each of the PI uptake values for the 387 isolates and controls and 'features' were the seven PI uptake curve parameters (*Supplementary file 4*). We then tested RF feature importance and showed with *variable importance plots* that the PI uptake parameters, AUC, $\mu^{max}$, and $r^{max}$ were the most informative features for the model (*Figure 4B*), consistent with our initial *InToxSa* assessment using JE2 and the *agrA* mutant (*Figure 1C, D*). Using the *proximity matrix* from the unsupervised RF model, we defined three main PI uptake clusters, corresponding with *low*, *moderate*, and *high* intracellular cytotoxicity categories. We labelled each of the 387 PI uptake data points with these three (*low*, *moderate*, and *high*) cytotoxicity categories and plotted the AUC and $r^{max}$ values against each other (*Figure 4C*). As expected, these parameters were strongly, positively correlated, suggesting that the AUC alone is sufficient to capture intracellular

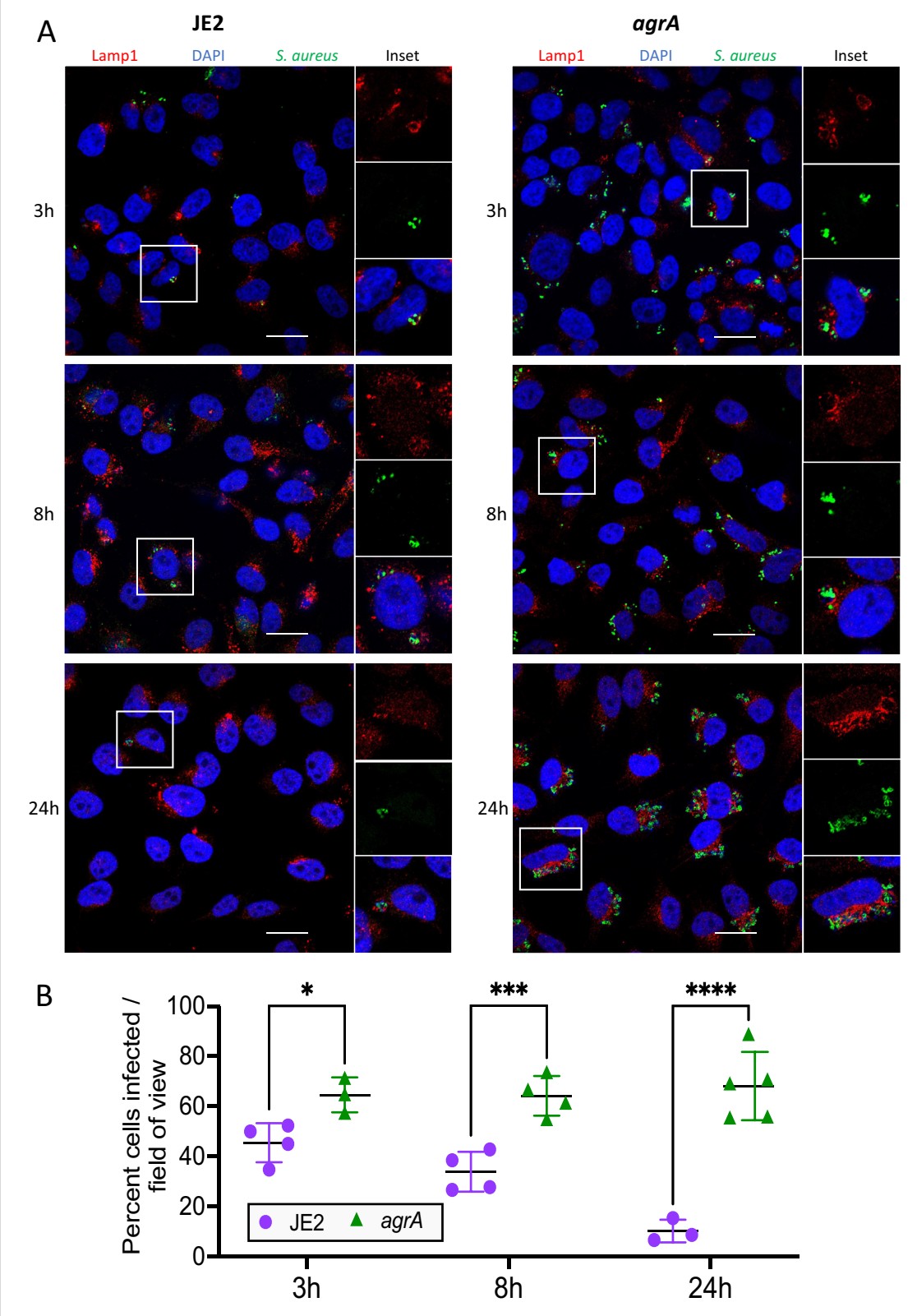

**Figure 2.** Fluorescence confocal microscopy of intracellular *S. aureus*. (**A**) HeLa cells were infected with *S. aureus* (wild-type JE2 or isogenic *agrA* mutant) and imaged at 3, 8, and 24 hr post-infection. Fixed cells were labelled with LAMP-1, *S. aureus* antibodies and 4',6-diamidino-2-phenylindole (DAPI). (**B**) Manual quantification of confocal microscopy. Graph shows the percentage of cells infected with *S. aureus* at each of the three timepoints. At least 50 cells (*n* cells = 51–112) were counted in 3–5 fields of view, with at least 12 cells counted per field (*n* field = 12–40). Shown are all data points,

*Figure 2 continued on next page*

*Figure 2 continued*

mean, and standard deviation. Significance was assessed using two-way analysis of variance (ANOVA). Null hypothesis (no difference between means) rejected for adj p < 0.05. *p = 0.04, ***p = 0.007, ****p = <0.0001.

cytotoxicity differences between *S. aureus* isolates. We used principal component analysis of the PI uptake data as an alternative unsupervised learning approach (*Figure 4—figure supplement 1*). When considering the first two components (67% of the variance explained), we observed a similar pattern where the same toxicity groups could be recognised within a cytotoxicity continuum among clinical isolates (*Figure 4—figure supplement 1*).

## GWAS analysis using *InToxSa* outputs to identify *S. aureus* genes linked to intracellular cytotoxicity

We next used GWAS to identify genetic correlates of strain-level cytotoxicity, expressed as mean PI uptake AUC. The fraction of cytotoxicity variation explained by genetic variation (heritability: $h^2$) was 49%, a figure lower than the ones obtained for other phenotypes such as vancomycin resistance (*Giulieri et al., 2022b*). A lower heritability could be resulting from the *InToxSa* assay variability or caused by differences in gene expression levels or due to epigenetic changes.

To assess the contribution of lineage effects relative to locus effects, we defined lineages using multidimensional scaling (MDS) of a pairwise genetic distance matrix generated by Mash, a tool that reduces genome content to a set of 'sketches' (hashed k-mers) (*Ondov et al., 2016*). Major MDS axes correlated with the most prevalent STs, for example ST239 was mainly defined by MDS1 (negative

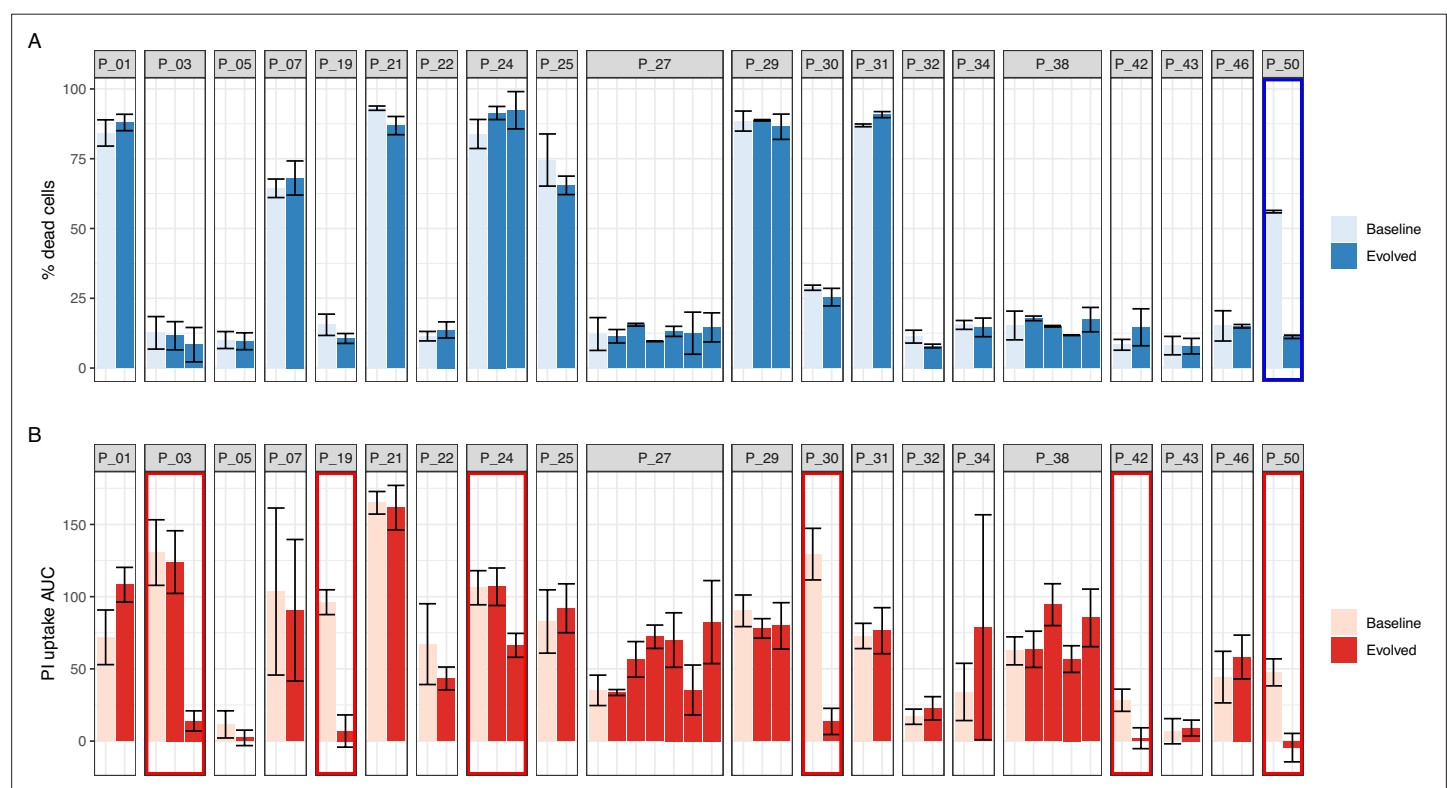

**Figure 3.** Performance of intracellular toxicity of *S. aureus* (*InToxSa*) against trypan blue exclusion assay. Comparative evaluation of *S. aureus* intracellular cytotoxicity with bacterial supernatants for 51 paired isolates from 20 patients with *S. aureus* bacteraemia. (**A**) Supernatant-based cytotoxicity on THP1 cells. Episode with significant difference in THP1 survival between baseline and evolved isolates is boxed in blue (p < 0.05). Assay performed in biological and technical duplicates. Bars represent mean percentage of dead cells; error bars show range between duplicates. Toxicity within isolate groups was compared using analysis of variance (ANOVA) with Bonferroni correction (**B**) propidium iodide (PI) uptake of infected HeLa cells. Values are mean area under the curve (AUC) and standard deviation. Episodes exhibiting significant phenotypic differences between baseline and evolved isolates are boxed in red (p < 0.05). Assay performed in biological and technical triplicates. PI uptake AUC within isolate groups was compared using ANOVA with Bonferroni correction.

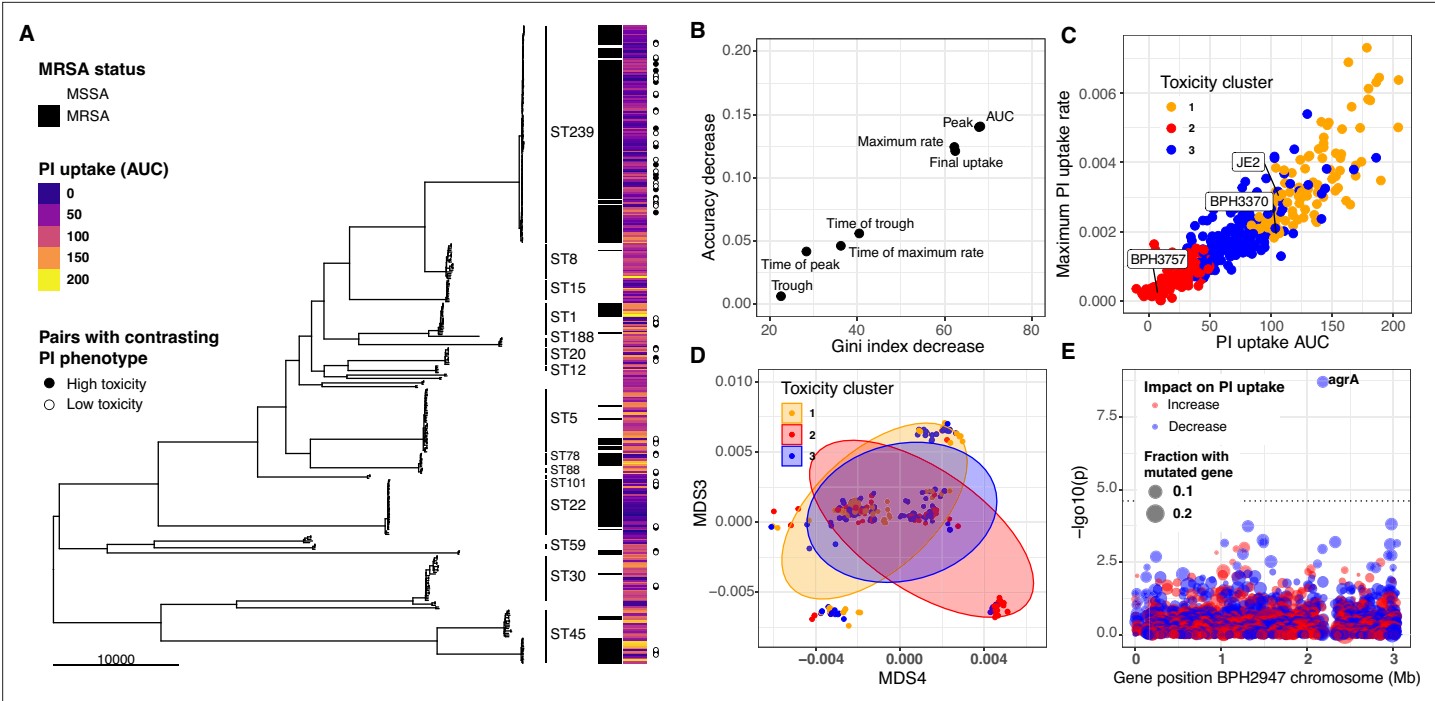

**Figure 4.** Intracellular cytotoxicity assessment of 387 bloodstream-associated clinical *S. aureus* isolates. (**A**) Maximum likelihood phylogeny based on 164,449 core-genome SNPs for 387 *S. aureus*, showing sequence type (ST) and MRSA distribution. The heatmap depicts the mean area under the curve (AUC) of cytotoxicity based on intracellular toxicity of *S. aureus* (*InToxSa*) propidium iodide (PI) uptake assay. AUC values range from non-cytotoxic (score: 0, dark blue) to highly cytotoxic (score: 200, yellow). Adjacent to the heatmap (closed and open circles) are 28 pairs of genetically related, but phenotypically discordant isolates (see *Figure 5*). (**B**) 'Variable importance plot' showing different PI uptake metrics (features) in an unsupervised random forest (RF) machine learning model. The higher the value of 'accuracy decrease' or 'Gini index decrease', the higher the importance of the feature in the model. (**C**) Scatter plot of the two most discriminatory PI uptake kinetic metrics (AUC and maximum PI uptake rate). Dots are coloured based on the clustering obtained from the proximity matrix of the RF model. (**D**) Scatter plot showing the two principal components with the strongest association with PI uptake (lineage effect as measured using pyseer). Dots and ellipses are coloured based on the clustering obtained from the proximity matrix of the unsupervised machine learning model. (**E**) Manhattan plot of gene-burden genome-wide association study (GWAS) of cytotoxicity (PI uptake AUC) of 387 clinical isolates.

The online version of this article includes the following figure supplement(s) for figure 4:

**Figure supplement 1.** Principal component analysis of mean value of propidium iodide (PI) uptake parameters.

**Figure supplement 2.** Lineage effects of cytotoxicity (area under the curve [AUC] of propidium iodide [PI] uptake).

**Figure supplement 3.** Interplay between multidimensional scaling (MDS) axes, sequence types (STs), and propidium iodide (PI) uptake.

**Figure supplement 4.** Genome-wide association study (GWAS) power calculation according to the number of isolates and number of variants for a phenotype with 50% heritability.

correlation) and MDS2 (positive correlation) (*Figure 4—figure supplement 3*). We then tested the association between the first 10 MDS axes (90% of the genetic variance explained) and the PI uptake phenotype in Pyseer (*Earle et al., 2016*; *Giulieri et al., 2022b*; *Lees et al., 2018*). In agreement with the initial observations based on the phylogeny and cytotoxicity heatmap (*Figure 4A*), we observed significant cytotoxicity-lineage associations represented by MDS3 and MDS4 (*Figure 4D*, *Figure 4—figure supplement 2*). Because of the ST-MDS lineages correlation, this is consistent with differences in cytotoxicity between clones (*Figure 4—figure supplement 3B*). Using the three cytotoxicity clusters defined by RF as categorical labels (*Figure 4C*), we plotted the 387 genomes along these two dimensions. While intracellular cytotoxicity was strongly associated with some *S. aureus* lineages, this analysis showed that lineage alone does not completely explain the phenotype, as indicated by the significant overlap between the three cytotoxicity clusters across MDS3 and MDS4 (*Figure 4D*). This pattern is consistent with other adaptive phenotypes (*Earle et al., 2016*; *Giulieri et al., 2022b*; *Su et al., 2021*) and suggests that locus effects from specific microevolutionary events modulate cytotoxicity, supporting the use of GWAS and convergent evolution approaches to identify these mutations.

Correcting for the observed population structure, we then used gene-burden GWAS to try and identify *S. aureus* loci significantly associated with intracellular cytotoxicity (PI uptake AUC) as a continuous variable. After correcting for multiple testing, only *agrA* reached the p < 0.05 significance threshold, supporting the important contribution of this locus to strain-level cytotoxicity (*Figure 4E*, *Supplementary file 5*). We also considered the highest ranking loci that did not reach genome-wide statistical significance. The second most significant gene, *secA2* (p = 1.5 × 10$^{-4}$) encodes the accessory ATPase to the Sec protein export system and is essential for the transport of SraP, a surface exposed and serine-rich staphylococcal protein which is associated with adhesion to, and invasion of epithelial cells and binding to human platelets (*Siboo et al., 2005*; *Yang et al., 2014*). Another high-ranking GWAS locus was *ileS* (p = 9.9 × 10$^{-4}$), encoding an isoleucyl-tRNA synthetase linked with mupirocin resistance, and previously associated with *S. aureus* cytotoxicity (*Yokoyama et al., 2018*).

## Identification of convergent mutations in genetic pairs with divergent *InToxSa* cytotoxicity profiles

Despite the relatively small sample size for this kind of analysis, the gene-burden GWAS detected the *agrA* locus with a high significance, but it did not have sufficient power to detect mutations other than those occurring in *agr* genes. This was expected with our sample size as demonstrated by our GWAS power calculations (*Figure 4—figure supplement 4*). Our power calculations have also showed that reducing the number of variants could substantially increase the yield of the analysis. This can be achieved by limiting the analysis to closely related isolates, with the added advantage of eliminating the bias related to the population structure. Therefore, we sought to identify rare mutations that might alter the intracellular cytotoxicity using comparative genomics approaches, a complementary strategy to microbial GWAS (*Chen and Shapiro, 2021*; *Giulieri et al., 2022b*; *Saund and Snitkin, 2020*). We used evolutionary convergence analysis to identify additional loci associated with intracellular cytotoxicity among the 387 *S. aureus* isolates. Our approach was to identify genetically related pairs of isolates with contrasting PI uptake AUC values from independent clades across the phylogeny and then search for homoplasic mutations between the pairs. We calculated genetic distances between all 387 genome-pairwise comparisons (149,769 combinations) and calculated a delta-PI uptake AUC value for each pair. We selected 28 phylogenetically independent *S. aureus* pairs (56 unique isolates) with a genetic distance <200 core-genome SNPs and a significant decrease in PI uptake AUC between reference (isolate-1) and control (isolate-2) (Wilcoxon rank-sum test) (*Figure 5A*). Variants within each pair (i.e., found in isolate-2 but not in isolate-1) were identified and annotated using a strategy that we have developed for *S. aureus* within-host evolution analysis (*Giulieri et al., 2022a*). We have previously shown that an SNP-calling approach using *de novo* assembly of one genome in a pair as a reference provided the most accurate estimate of the genetic distance (*Higgs et al., 2022*). There were between 0 and 206 mutations within the 28 pairs (*Figure 5B*). Mapping the genes in which these mutations were found back to a core-genome phylogeny constructed from the 56 paired *S. aureus* genomes showed convergent (homoplasic) mutations in *agrA*, supporting the GWAS findings. In addition, our analysis identified potentially convergent mutations in several other genes (6 with three independent mutations and 35 with two independent mutations) (*Figure 5C–E*, *Supplementary file 6*). However, because of the strong lineage effect and the paucity of representation for some *S. aureus* lineages (clonal complexes [CC] and STs), half of these mutations were only found in ST239, a well-represented lineage in our collection. In addition to target loci dedicated to the regulation of virulence factors such as the *agr* locus or involved in adhesion to host extracellular matrix proteins such as fibronectin and elastin (*fnbA* and *ebpS*), some of the convergent mutations were found in genes involved in metabolic processes (*ribA*, *purF*, *sbnF*, *ilvB*, *lysA*, and *araB*), associated with the cell wall (*fmtB*), devoted to the last step of the cell wall teichoic acid biosynthesis (*tarL'*), implicated in DNA repair (*mfd*), in protein transport (*secA2*), in solute transport (*glcA*, *opp-3A*, and *thiW*), in respiration (*cydA*) and found in a phage-associated locus (SAUSA300_1930). Aside from *agrA* and *agrC* genes (*Giulieri et al., 2018*; *Laabei et al., 2014*; *Laabei et al., 2015*; *Mairpady Shambat et al., 2016*), and those found in the promoter of the *tar* locus (*Brignoli et al., 2022*; *Laabei et al., 2014*), mutations in the other loci have not been associated with the reduction of cytotoxicity in clinical *S. aureus* isolates. Interestingly, homoplasic mutations were also found in the gene *ausA*, known to be involved in *S. aureus* escape from epithelial cell endosomes and the phagosome of phagocytic cells (*Blättner et al., 2016*).

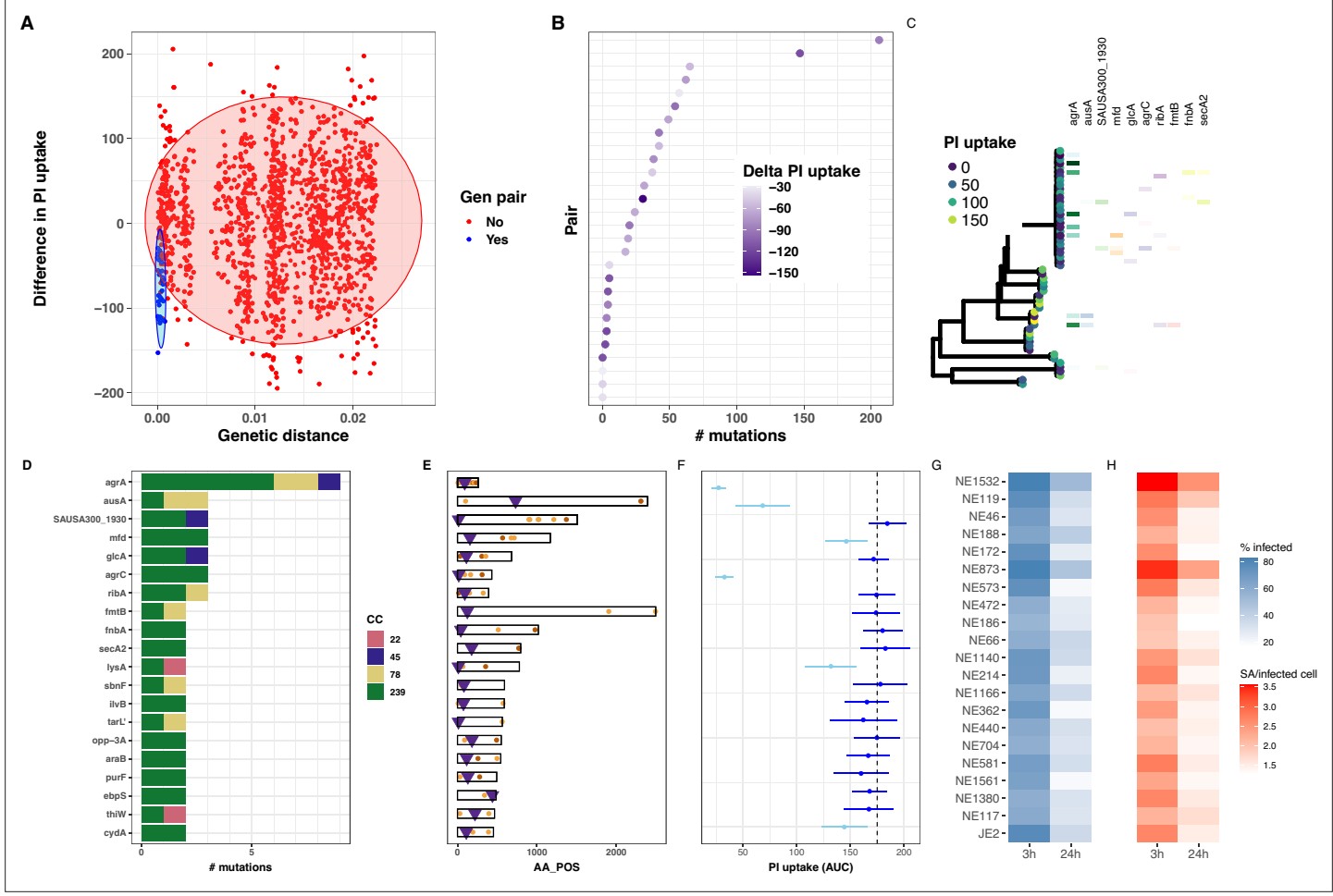

**Figure 5.** Evolutionary convergence analysis to identify *S.aureus* genes linked with intracellular cytotoxicity. (**A**) Distribution of genetic distance determined by pairwise comparisons using MASH distances among the 387 *S. aureus* genomes, against the difference in propidium iodide (PI) uptake area under the curve (AUC) between each pair. The shaded circles denote the 95% multivariate *t*-distribution (blue: pairs included in the convergence analysis; red: pairs excluded from the analysis). (**B**) Ranked distribution of the difference in PI uptake AUC between the 28 pairs. Heatmap shows reduction in AUC values. (**C**) Core-genome phylogeny for the 28 pairs of isolates. Tree tips are coloured by PI uptake AUC. Aligned with the phylogeny, the 10 first genes targeted by convergent mutations are shown. (**D**) Number of mutations detected for each of the 20 genes, coloured by *S. aureus* ST. (**E**) Location of convergent mutations in each gene (non-synonymous in orange, truncating in maroon), purple triangles indicate the position of transposon inserted in tested transposon mutant. (**F**) Effect of loss of function for each of the 20 genes on intracellular cytotoxicity measured by intracellular toxicity of S. aureus (*InToxSa*), using mutants from the Nebraska transposon library. Dotted line shows mean PI uptake AUC of positive control strain JE2. Depicted are mean (dot) and standard deviation (SD; bar) of biological triplicates. Mutants causing significantly lower PI uptake AUC to JE2 are depicted in light blue, non-significant changes are in dark blue (Wilcoxon rank-sum test, corrected for multiple testing). (**G, H**) Operetta high-content imaging analysis for each of the 20 Nebraska transposon mutants and JE2 positive control. Heatmaps show the percentage of HeLa cells infected with each transposon mutant (blue) and the number of bacteria per infected cells at 3 and 24 hr post-infection (red).

The online version of this article includes the following figure supplement(s) for figure 5:

**Figure supplement 1.** Identification of convergent mutations associated with intracellular toxicity and persistence.

**Figure supplement 2.** Automated genetic pairs analysis (100 replicated sets of phylogenetically independent genetic pairs).

**Figure supplement 3.** Scatter plots showing the statistically significant inverse Pearson correlations of (**A**) propidium iodide (PI) area under the curve (AUC) with the percentage of infected cells; and (**B**) PI AUC with the number of *S. aureus* per infected cells at 24 hr post-infection.

We focussed our analysis on genetic pairs with loss of cytotoxicity, to be consistent with the hypothesis that adaptive mutations arising during infection would promote a low-toxicity phenotype as previously shown (*Giulieri et al., 2018*; *Laabei et al., 2014*). This hypothesis is supported by our gene-burden GWAS, where 10 out of 10 most significant hits are associated with loss of cytotoxicity (mean normalised PI AUC decrease = −0.8, *Supplementary file 5* and *Figure 4E*). To explore whether including pairs with toxicity gain would alter the analysis, we selected four additional phylogenetically

independent pairs with gain of toxicity. Adding these pairs did not reveal new genes with convergent mutations, however, this analysis revealed an additional homoplasic mutation in *ausA* (*Figure 5—figure supplement 1*).

To further confirm the roles of genes bearing homoplasic mutations in pairs with discordant phenotype, we compared PI uptake between pairs with and without homoplasic mutations. We ran the analysis in 100 replicates of independent pairs combinations and assessed the associations between within-pairs mutations (aggregated at the gene level) and PI uptake difference within the pair using linear regression. For 6 out of the 20 genes originally identified in the genetic pairs analysis (including *agrA*, *glcA*, *ribA*, *fmtB*, *sbnF*, and *tarL'*), this analysis showed that for >95% of replicates, the regression coefficient was above 0, thus providing a robust statistical support for our findings (*Figure 5—figure supplement 2*). For the gene *ausA*, 86% of replicates had a beta value above 0, however 78% of *agr*C regression replicates had a beta value below 0.

## Functional assessment of genes with convergent mutations

To assess the functional consequences of the convergent mutations (caused by at least two homoplasic mutations per gene), we again turned to the Nebraska transposon library and selected transposon mutants for 20 genes we had identified (*Figure 5D*). We used *InToxSa* to assess the effect of gene disruption on the intracellular cytotoxicity phenotype for each mutant compared to the JE2 wild type (*Figure 5F*). Over the 20 transposon mutants tested, six showed a statistically significant reduction in cytotoxicity, namely NE1532 (*agrA*), NE119 (*ausA*), NE188 (*mfd*), NE873 (*agrC*), NE1140 (*lysA*), and NE117 (*cydA*). Strains with transposon insertions in *agrA*, *agrC*, and *ausA* showed a highly significant reduction in PI uptake AUC (adjusted p = $5.4 \times 10^{-4}$, Wilcoxon rank-sum test), confirming their reported roles in affecting bacterial cytotoxicity and validating our convergence analysis (*Figure 5F*; *Blättner et al., 2016*; *Das et al., 2016*; *Laabei et al., 2021*; *Mairpady Shambat et al., 2016*). We extended this analysis and used high-content, high-throughput microscopy to observe and quantify in an unbiased manner the impact of each mutation on the *S. aureus* infectivity and intracellular persistence (see methods). There was an inverse relationship between *InToxSa* and high-content imaging outputs, with the three mutants most reduced in cytotoxicity showing both a higher percentage of infected cells recovered after 24 hr of infection, and a high number of bacteria per infected cell at 24 hr post-infection as compared to the wild-type control JE2 (*Figure 5G, H – Figure 5—figure supplement 3*).

## Functional assessment of specific convergent mutations

To further assess the impact of specific convergent mutations on intracellular cytotoxicity, we used site-directed mutagenesis in *S. aureus* BPH3370 (ST239) to recreate a subset of the convergent mutations. We selected isolate BPH3370 for these experiments as it displayed high *InToxSa* PI uptake AUC (comparable to JE2, *Figure 4C*) cytotoxicity without bearing any of the convergent mutations we intended to introduce. We focussed our attention on mutations likely to affect protein function and based on the attenuation in cytotoxicity of the cognate transposon mutants. We selected six mutations, previously not documented nor characterised, including non-synonymous mutations leading to residue substitution (*agrA* E7K and *cydA* R390C [a reversion of C390R]), frameshifts leading to truncated proteins (*agrC* G310 frameshift, *ausA* K2308 frameshift, and *lysA* K354 frameshift), and introduction of a stop codon (*mfd* W568 stop codon) in the sequences of convergent genes (*Figure 6A* and *Figure 6—figure supplement 1*). We then used *InToxSa* to assess the cytotoxicity of each targeted mutant, compared to BPH3370 wild type, JE2 and the corresponding JE2 Nebraska transposon mutant, for each of the six loci (*Figure 6B*, *Figure 6—figure supplement 1*). We observed that recreation of the E7K *agrA* mutation, the *agrC* G310 frameshift mutation and the *ausA* K2308 frameshift mutations lead to a significant reduction in intracellular cytotoxicity in BPH3370 (*Figure 6B*). However, the W568 stop codon *mfd* mutation, the K354 frameshift *lysA* mutation, and the *cydA* R390C mutation had no significant effect on the cytotoxicity of the BPH3370 strain (*Figure 6—figure supplement 1*). It is noteworthy that transposon insertions in these three genes also had a less pronounced effect on the phenotype of JE2 strain as compared to the *agr* and *ausA* loci (*Figure 5F*).

We predicted that the *ausA* K2308 frameshift causing a 11 base pair deletion mutation in BPH3370 would lead to a loss of aureusimine biosynthesis. This was because the frameshift occurred within the *ausA* reducing domain and would thus prevent the release of the dipeptide L-Val-L-TyrT2 to form the intermediate amino aldehyde, with no cyclisation to form the imine (*Figure 6C*; *Zimmermann and*

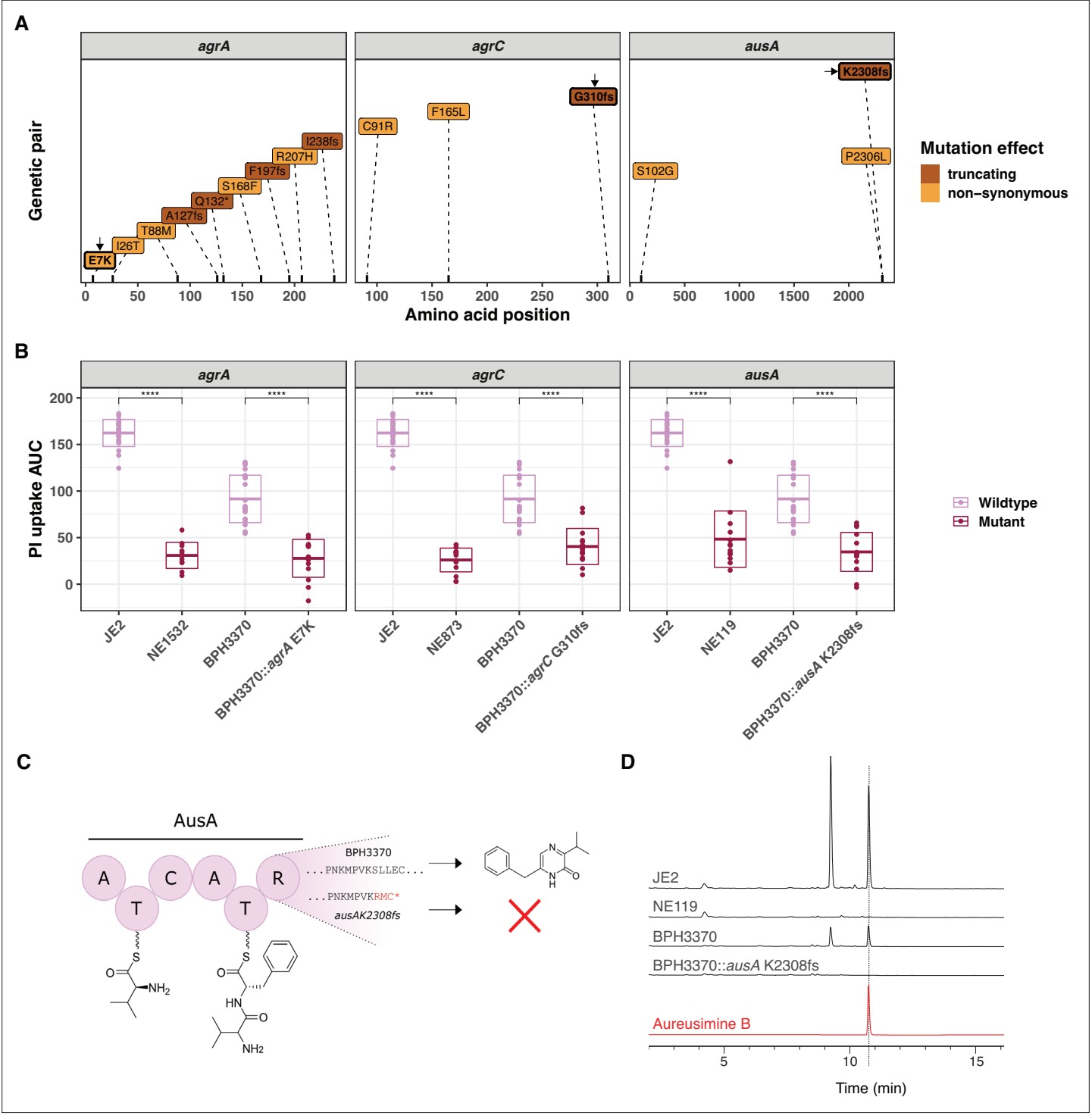

**Figure 6.** Introduction of convergent *agrA*, *agrC*, and *ausA* mutations in the clinical isolate BPH3370 reduces its intracellular cytotoxicity while *ausA* mutation affects aureusimine B production. (**A**) Position and nature of convergent mutations identified in the genes *agrA*, *agrC*, and *ausA*. For each gene, the amino acid position affected by mutations is shown on the *x*-axis for each gene. Convergent mutations causing a significant contrasting propidium iodide (PI) uptake phenotype are coloured according to their consequence on protein function: non-synonymous (orange), truncating characterised by the introduction of a frameshift (fs) or a stop codon (*) (maroon). (**B**) Effect of convergent mutations on the intracellular cytotoxicity of the clinical isolate BPH3370. The PI uptake area under the curve (AUC) values for JE2, the cognate Nebraska transposon mutants of convergent genes, BPH3370 wild type and BPH3370 bearing the mutations affecting *agrA*, *agrC*, and *ausA*. The crossbar represents mean and standard deviation (p < 0.0001). (**C**) Predicted impact of K2308 frameshift mutation (K2308fs) on aureusimines. (**D**) HPLC analysis of *S. aureus* ethyl-acetate extracts for aureusimines compared to an Aureusimine B synthetic standard.

*Figure 6 continued on next page*

*Figure 6 continued*

The online version of this article includes the following figure supplement(s) for figure 6:

**Figure supplement 1.** Introduction of convergent *cydA*, *lysA*, and *mfd* mutations in the clinical isolate BPH3370 and intracellullar cytotoxicity (*InToxSa*) assessment.

*Fischbach, 2010*). As expected, high-performance liquid chromatography (HPLC) analysis confirmed the absence of aureusimines in the BPH3370 K2308 frameshift mutant, similar to the transposon *ausA* mutant (NE119) (*Figure 6D*).

## Discussion

Recurrent and persistent staphylococcal infections have been proposed to result from within-host selective pressures leading to the evolution of adaptive traits by the bacteria, a process also observed in other human bacterial pathogens (*Didelot et al., 2016*; *Gatt and Margalit, 2021*; *Giulieri et al., 2022a*). The emergence of mutations affecting regulators controlling toxin production has been proposed as a mechanism enabling *S. aureus* to adapt to its host while evading cellular immune responses (*Giulieri et al., 2022a*; *Young et al., 2012*; *Young et al., 2017*). Identifying the molecular signatures supporting the pathoadaptation of *S. aureus* at the host cell interface is important for understanding how *S. aureus* can cause persistent, difficult-to-treat infections lasting many months (*Gao et al., 2015*).

Several studies of *S. aureus* clinical isolates have attempted to identify such signatures by assessing the cytotoxicity of bacterial supernatants applied onto host cells, in an *ex-cellulo* fashion (*Giulieri et al., 2018*; *McConville et al., 2022*; *Recker et al., 2017*). Such assessments would be adequate if *S. aureus* was an extracellular pathogen exerting its cytotoxicity from without host cells (*Soe et al., 2021*), but *S. aureus* is a facultative intracellular pathogen able to invade and persist in a wide range of eukaryotic cells (*Al Kindi et al., 2019*; *Krauss et al., 2019*; *Luqman et al., 2019*; *Musilova et al., 2019*; *Sinha and Fraunholz, 2010*). We developed the *InToxSa* cytotoxicity assay to address this shortcoming and to try and identify *S. aureus* pathoadaptive mutations that support a *S. aureus* intracellular lifestyle. Moreover, the *InToxSa* assay provides a maximum throughput of 7× 96-well plates per week, thus corresponding to 98 distinct clinical strains testable per week (encompassing 6 individual replicates, each tested across 2 different days/plates) which could be upscaled 4-fold with the adoption of a 384-well plate format. By harnessing the power of comparative and statistical bacterial genomics with *InToxSa* readouts for a large collection of bacteraemia-associated *S. aureus* isolates, we identified mutations in *S. aureus* that reduced the intracellular cytotoxicity and increased intracellular persistence.

We showed the performance and sensitivity of *InToxSa* with the identification of cytotoxicity differences between *S. aureus* isolates that had not previously been detected by *ex-cellulo* methods (*Figure 3*; *Giulieri et al., 2018*). The difference in phenotypic outputs for both methods may be in part explained by the different cell types exploited for the readout (THP1 macrophages for trypan blue exclusion assay versus HeLa-CCL2 epithelial cells for *InToxSa*) and the bacterial fraction examined (culture supernatants versus bacterial cells) (*Figure 3*). The capacity of *InToxSa* to detect subtle phenotypes missed by gross cytotoxicity assessments is also conferred by its temporally granular and objective measurements of PI uptake as a marker of host cell viability. *InToxSa* assesses the *S. aureus* toxicity caused by bacterial virulence factors produced in response to the intracellular environment and is proportional to a defined bacterial load. This approach contrasts with methods relying on the presence of toxins accumulating over time in bacterial supernatants and whose production relies almost solely on the functionality of the Agr quorum sensing system (*Altman et al., 2018*; *Giulieri et al., 2018*; *McConville et al., 2022*).

We further showed the performance of this analytical pipeline by readily identifying mutations in loci such as the Agr quorum sensing system, which is well known to control *S. aureus* cytolytic activity (*Giulieri et al., 2018*; *Giulieri et al., 2022a*; *Laabei et al., 2021*; *Mairpady Shambat et al., 2016*; *Recker et al., 2017*). However, our approach also enabled discovery of mutations in less characterised systems, including changes in *ausA*, that reduced *S. aureus* cytotoxicity and increased intracellular persistence of clinical isolates. AusA is a non-ribosomal peptide synthetase responsible for production of aureusimines, pyrazinone secondary metabolites. Our observations are consistent with previous

reports showing that aureusimines contribute to the phagosomal escape of *S. aureus* JE2 to the cytosol (*Blättner et al., 2016*; *Wilson et al., 2013*).

Interestingly, *S. aureus* mutants that were most affected in cytotoxicity also had a propensity to persist intracellularly (*Figure 5*). Infected host cells have been proposed as Trojan horses for intracellular *S. aureus*, increasing the risks of systemic dissemination to organs, such as the liver and kidneys, following bacteraemia and contribute to infection persistence (*Jorch et al., 2019*; *Surewaard et al., 2016*; *Thwaites and Gant, 2011*). Whilst *agr*-dysfunctional isolates were associated with persistent infections (*Fowler et al., 2004*; *Schweizer et al., 2011*), intracellular persistence caused by mutations in *agr* loci could possibly constitute a population bottleneck (*Pollitt et al., 2018*; *Spaan et al., 2013*). However, such population bottlenecks may be transient as it has been suggested that mutations arising in *agr* defective pathoadapted clinical isolates could possibly compensate for the loss of *agr* functionality and restore *S. aureus* virulence, suggesting a stepwise within-host evolution of clinical isolates (*Altman et al., 2018*; *Giulieri et al., 2022a*).

Current statistical genomics strategies in human genetics support combining allele-counting methods (GWAS), for the detection of common variants, with comparative genomics approaches to identify rare variants (*Singh et al., 2022*; *Trubetskoy et al., 2022*). In microbial genomics, this strategy is best achieved by combining microbial GWAS and convergent evolution studies (*Chen and Shapiro, 2021*; *Giulieri et al., 2022b*; *Giulieri et al., 2018*;(*Guérillot et al., 2018*) *Saund and Snitkin, 2020*). Whilst our GWAS approach only identified *agrA* as significantly associated with low cytotoxicity (*Figure 4E*), our evolutionary convergence analysis on genetic pairs among our 387 bacteraemia isolates identified mutations in several *S. aureus* genes that led to reduced cytotoxicity (*Figure 5*). However, only convergent mutations occurring in *agrA*, *agrC*, and *ausA* were confirmed to affect the cytotoxicity and intracellular persistence phenotypes when introduced into a clinical isolate (*Figure 6*). This may be due to epistatic effects or combinations of mutations within a specific *S. aureus* strain may be acting in concert to control the expression of the numerous bacterial cytolytic determinants, underscoring the need to functionally confirm the findings of the convergence analysis. We acknowledge as a limitation of the convergent evolution analysis its reliance on a set of phylogenetic independent pairs, selected by the investigator. Thus, the choice of a different set of pairs may lead to a different list of homoplasic changes. To overcome this limitation, we have performed a repeated pair sampling and assessed the association between homoplasic mutations in PI changes. This approach provided robust support to 6 out of 20 findings of the convergent analysis (including *agrA* and *ausA*).

Our approach to integrate GWAS with convergence analyses underscores an important point regarding the choice of samples in genotype–phenotype studies. For future studies, it would be ideal to enrich for genetically related isolates, for example, isolated from the same patient or from a clonal outbreak, if logistically feasible. This would reduce the genetic distance and could therefore increase the yield of the analysis.

Our study also shows that intracellular cytotoxicity levels vary between STs. Despite causing bacteraemia, the ST22 and ST239 isolates were overall less cytotoxic than the ST8 isolates in our collection (as shown on the heatmap in *Figure 4A*), further corroborated by the direct cytotoxicity comparison between strains JE2 (ST8) and BPH3370 (ST239) (*Figure 6B*). The evolution of reduced Agr functionality (and thus cytotoxicity) in hospital-acquired ST239 and ST22 isolates has already been reported by our group and others and is confirmed by the *InToxSa* outputs (*Baines et al., 2015*; *Collins et al., 2008*; *Giulieri et al., 2018*; *Laabei et al., 2021*; *Li et al., 2016*). Consistent with their reduced cytotoxicity and with our hypothesis of inverse correlation between toxicity and intracellular replication, ST239 isolates caused higher degrees of bacterial persistence in infected animal models (*Baines et al., 2015*; *Li et al., 2016*) and showed increased intracellular persistence in osteoblasts (*Bongiorno et al., 2021*). Within the limits of our experimental settings, the relatively lower cytotoxicity of ST239 and ST22 isolates indicates that the amplitude of this phenotype should probably be considered within a genetic lineage. The inclusion of representative isolates per lineage, with defined cytotoxicity levels, would identify cytotoxicity thresholds and perhaps allowing identification of more subtle genomic changes affecting phenotypes. Moreover, some of the loci detected by the convergence and GWAS analyses may also have more pronounced effects in some lineages than in others. For instance, mutations affecting *tarL* and *secA2* may affect the export and secretion of virulence factors that are only present in a subset of lineages, thus explaining the absence of effect on cytotoxicity caused by the cognate transposon mutants (*Figure 5F*).

We developed *InToxSa* using HeLa cells, a well-defined, adherent, and non-phagocytic cellular model (*Das et al., 2016*; *Stelzner et al., 2020a*; *Stelzner et al., 2020b*). We used adherent epithelial cells because they can be maintained for extended infection periods and so allow the acquisition of useful kinetic measurements of cytotoxicity. However, we also acknowledge the limitation in using these cells in that they do not have the bactericidal modalities of the phagocytes encountered by *S. aureus* in the bloodstream (*Brinkmann et al., 2004*; *Chow et al., 2020*; *Krause et al., 2019*). Neutrophils are among the first immune cells to engage *S. aureus* during bacteraemia (*Brinkmann et al., 2004*). However, neutrophils have a relatively short *in vitro* lifespan following their purification from blood and would not be well suited to an *InToxSa*-style assay format (*Ge et al., 2020*; *Rosales, 2020*; *Tak et al., 2013*; *Zwack et al., 2022*). Polymorphonuclear cell lines such as HL-60, exploited in other *ex-cellulo* assays, may represent an alternative to primary neutrophils (*McConville et al., 2022*; *Rose et al., 2015*). These cells display some of the same important biological functions as neutrophils, including neutrophil extracellular traps (*Scieszka et al., 2020*), critical in the clearing of *S. aureus* (*Brinkmann et al., 2004*; *Greenlee-Wacker et al., 2014*; *Zwack et al., 2022*). While PI uptake by these cells could be used as a readout of their viability, HL-60 cells also do not cover all the bactericidal enzymatic activities of primary neutrophils, a potential limitation for their use (*Nordenfelt et al., 2009*; *Yaseen et al., 2017*).

We used *InToxSa* to identify *S. aureus* pathoadaptive mutations, enriched in bacterial populations that are associated with human disease (e.g., upon transit from colonising to invasive). We hypothesised that these mutations would support an intracellular persistence for *S. aureus*. Our future research will focus on understanding how these genetic changes might be allowing the bacterium to avoid cell-intrinsic surveillance systems, such as lytic programmed cell death; the self-destructive processes restricting systemic progression of intracellular bacterial pathogens (*Wanford et al., 2022*). Unlike well-described intracellular Gram-negative bacteria, *S. aureus* does not have effector proteins to block lytic programmed cell death (*Soe et al., 2021*). Pathoadaptive mutations such as those arising in the *agr* locus might prevent cellular injuries caused by *S. aureus* toxins under Agr control, that would be sensed by cell-intrinsic surveillance platforms such as the inflammasomes and trigger cell death (*Krause et al., 2019*; *Soe et al., 2021*). Loss-of-function mutations in *ausA*, preventing the biosynthesis of aureusimines might be confining *S. aureus* to a lysosomal compartment where the bacteria have the potential to replicate, and conceivably evade host surveillance mechanisms (*Blättner et al., 2016*; *Flannagan et al., 2016*; *Grosz et al., 2014*; *Moldovan and Fraunholz, 2019*).

## Conclusion

Current large-scale comparative genomics of *S. aureus* bacteraemia isolates can be further refined by including underexplored pathogenicity traits such as the capacity of *S. aureus* to invade and survive in host cells. We have addressed this poorly characterised trait of *S. aureus* pathogenicity by creating the *InToxSa* assay that measures the intracellular cytotoxicity of many hundreds of *S. aureus* clinical isolates at scale. We showcase the robustness and reproducibility of phenotypic outputs which, in combination with comparative and statistical genomics, have confidently identified convergent mutations arising in *agr* and *ausA* genes that reduced the intracellular cytotoxicity and increased the intracellular persistence of bacteraemia isolates during infection. The adoption of the *InToxSa* methodology in future pheno-genomics studies would improve the detection of pathoadaptive mutations supporting the persistence and relapse of *S. aureus* infections.

## Materials and methods

**Key resources table**

| Reagent type (species) or resource | Designation | Source or reference | Identifiers | Additional information |
|---|---|---|---|---|
| Gene (*S. aureus*) | BPH2947 | GenBank | Accession GCF_900620245.1 | |
| Strain, strain background (*Escherichia coli*) | *E. coli* strain IM08B | *Monk et al., 2015* | IM08B | Electrocompetent cells |
| Cell line (*Homo sapiens*) | Epithelial cells | ATCC | HeLa-CCL2 | |

*Continued on next page*

*Continued*

| Reagent type (species) or resource | Designation | Source or reference | Identifiers | Additional information |
|---|---|---|---|---|
| Chemical compound, drug | Antibiotic | Baxter | Gentamicin | 80 and 40 mg/ml |
| Recombinant DNA reagent | Antibiotic | Ambi | Lysostaphin | 10 mg/ml |
| Chemical compound, drug | Nucleic marker | Sigma | Propidium iodide | 1 mg/ml |
| Antibody | Anti-*S. aureus* (Rabbit polyclonal) | WEHI-antibody technology platform | Customised | IF (1:1000) |
| Antibody | Anti-rabbit (Donkey polyclonal) coupled to Alexa Fluor 488 | Thermo Fisher-Invitrogen | Cat#: A-21206 | IF (1:2000) |
| Other | Phalloidin-TRITC | Sigma | P1951 | IF (1:4000) |
| Other | DAPI | Sigma-Merck | 10236276001 | IF (1:2000) |

## *S. aureus* isolates

Clinical isolates were selected from a combined collection of 843 clinical isolates of *S. aureus* bacteraemia (*Giulieri et al., 2018*) that was obtained from the vancomycin substudies of the Australian and New Zealand Cooperative on Outcome in Staphylococcal Sepsis (ANZCOSS) study (*Holmes et al., 2011*) and the Vancomycin Efficacy in Staphylococcal Sepsis in Australasia (VANESSA) study (*Holmes et al., 2018*). We selected 387 isolates to maximise the likelihood to detect phenotype–genotype associations by sampling different lineages and enriching for episodes where multiple isolates per patient were available (see *Supplementary file 3*).

## Whole-genome sequencing

After subculturing strains twice from −80°C glycerol stock, DNA was extracted using the Janus automated workstation (PerkinElmer) or manual extraction kits (Invitrogen PureLink genomic DNA kit or the Sigma GenElute kit). Normalised DNA (at a concentration of 0.2 ng/ml) was prepared for sequencing using Nextera XT DNA (Illumina) and sequencing was performed on Illumina MiSeq and NextSeq platforms. Reads quality was assessed based on mean read depth and percentage of *S. aureus* reads as computed using Kraken2 (*Wood et al., 2019*). Reads were assembled using Shovill, an assembly pipeline that optimises the Spades assembler (https://github.com/tseemann/shovill) (*Bankevich et al., 2012*). Annotation was performed using Prokka, with a minimal contig size of 500 bp (*Seemann, 2014*). Assembly and annotation metrics were used for further quality control of the reads. Genetic distance between clinical isolates was calculated using Mash with a sketch size of 10,000 (*Ondov et al., 2016*). We used the distance matrix generated by Mash to perform MDS using the function 'cmdscale()' in base R. Multi-locus ST were inferred from the assemblies using mlst (https://github.com/tseemann/mlst) (*Seemann, 2020a*). We assessed the correlation between the most prevalent ST and the MDS axes using the get_correlations() function in the R package bugwas (*Earle et al., 2016*).

## Variants calling: single reference

Clinical isolates' reads were mapped to internal reference BPH2947 (accession GCF_900620245.1), an ST 239 reference genome that was generated from the collection. We used snippy, v4.6.0 for mapping and variant calling, with default settings (https://github.com/tseemann/snippy) (*Seemann, 2020b*). The core-genome alignment was constructed using Snippy-core. We defined core genome as positions where at least 90% of the sequences had a minimum coverage of 10 reads and used Goalign v0.3.4 and SNP-sites v2.5.1 to extract core-genome positions. To infer a maximum likelihood phylogenetic tree of the clinical isolates collection we ran IQ-TREE v2.0.3 using a GTR-G4 model. We used HomoplasyFinder (*Crispell et al., 2019*) to identify homoplasic sites based on the consistency index. The consistency index was calculated with the following formula: (Number of nucleotides at site − 1)/ Minimum number of changes on tree.

## Construction of mutants by allelic exchange

Engineering of convergent mutations in *agrA*, *agrC*, *ausA*, *mfd*, *lysA*, and *cydA* genes in the strain BPH3370 was performed by allelic exchange as described previously (*Monk and Stinear, 2021*) using oligonucleotides described in *Brinkmann et al., 2004*; *Supplementary file 7* (outlining residues modified by convergent mutations). Upstream and downstream regions of each mutation were PCR amplified and gel extracted and then a splice by overlap extension PCR was performed to generate each insert. Each insert was cloned into linearised pIMAY-Z vector by Seamless Ligation Cloning Extract (SLiCE) cloning (*Zhang et al., 2012*) to generate six plasmids. Each plasmid was separately transformed into *E. coli* strain IM08B (*Monk et al., 2015*) confirmed by colony PCR, then purified and transformed into *S. aureus* strain BPH3370 by electroporation. Mutant candidates were screened by Sanger Sequencing (Australian Genome Research Facility, Melbourne, VIC, AUS) and positive clones were validated by whole-genome sequencing on an Illumina Miseq or NextSeq550 platforms (Illumina, San Diego, CA, USA) to confirm their genotype. The resultant reads were mapped to the BPH3370 reference genome and mutations were identified using snippy (v4.6.0, https://github.com/tseemann/snippy) (*Crispell et al., 2019*).

## Clinical isolates library preparation

The collection of clinical isolates was prepared to be readily inoculated from 96-well microtitre plates. Clinical isolates were grown in 10 ml Brain Heart Infusion (BHI) broth (BD Bacto) from single colonies to stationary phase. Briefly, a volume corresponding to 1-unit $OD_{600}$ for each culture was centrifuged at $10,000 \times g$ for 5 min. The bacterial pellets were washed once with 500 µl of fresh BHI and centrifuged again. The washed bacterial pellets were resuspended in 600 µl of storage media (BHI containing 40% glycerol), vortexed briefly and 200 µl were distributed across 96-well microtitre plates. To prevent operator and plate effect biases, the 387 isolates were randomly distributed with each plate to include 29 distinct isolates, represented in non-contiguous technical triplicates. Built-in controls for cytotoxicity were included in each plate. The wild-type JE2 strain was used as positive cytotoxicity control and the BPH3757 strain, an ST239 isolate bearing the T88M *agrA* mutation described in *Giulieri et al., 2018*, as a non-cytotoxic control. Six wells were kept empty to monitor the viability of non-infected controls and account for residual PI uptake. Plates were stored at −80°C. Each plate was at least tested in three biological replicates.

## Tissue culture

HeLa-CCL2 cells (sourced from the ATCC) were maintained and propagated in Dubelcco's modified Eagle medium (DMEM) + GlutaMAX (4.5 g/l D-glucose and 110 mg/l sodium pyruvate) supplemented with heat-inactivated 10% foetal bovine serum (Gibco) and in absence of antibiotics. The cells were regularly tested for the presence of mycoplasma.

## InToxSa assay

*S. aureus* isolates were inoculated directly from stabbed frozen parsed plates stock into 100 µl of BHI broth dispatched in flat bottom 96-well microtitre plates. Inoculated plates were incubated for 16 hr in a heat-controlled plate reader (CLARIOstar plate reader, BMG Labtech) set at 37°C. Bacterial growth was assessed by $OD_{600}$ measurement every 10 min. The endpoint optical densities of cultures were used to infer bacterial density (1-unit $OD_{600}$ corresponding to $5 \times 10^8$ bacteria/ml). Bacterial cultures were standardised and serially diluted in DMEM to reach an MOI 10. 100 µl of bacterial suspension were added to infect 40,000 HeLa-CCL2 cells grown (70% confluence per well) in 96-well black plates, clear bottom (Sigma). Infection was synchronised by centrifugation at $500 \times g$ for 10 min (Eppendorf 5810R) at room temperature. Infected plates were incubated 2 hr at 37°C and 5% $CO_2$ to allow for *S. aureus* internalisation. The infective media was then discarded, and cells washed once with sterile phosphate-buffered saline (PBS) and further incubated 1 hr with 100 µl DMEM containing cell impermeable antibiotics (80 µg/ml gentamicin [Baxter] and 10 µg/ml of lysostaphin [Ambi]) at 37°C and 5% $CO_2$ (*Kim et al., 2019*). This first step of antibiotic-protection assay was followed by another using a lower antibiotic concentration (40 µg/ml gentamicin and 10 µg/ml lysostaphin), in media supplemented with 5% fetal bovine serum (Gibco), and 1 µg/ml PI, a live cell-impermeant nucleic acid dye (Sigma). Plates were then incubated in the CLARIOstar Plus plate reader (BMG Labtech) set at 37°C and 5% $CO_2$ throughout the infection (up to 20 hr post-infection). The fluorescence signal emitted by

PI-positive cells was acquired every 6 min from each well (excitation at 535 nm, emission at 617 nm, using the spiral well scanning mode with 50 flashes per well). Non-infected control cells were permeabilised with 0.1% Triton X-100 to determine the maximum level PI uptake and HeLa cell death.

## High-content imaging

The Operetta high-content microscope (PerkinElmer) was employed to accurately quantify and analyse intracellular persistence at the single-cell resolution. HeLa-CCL2 cells were seeded in Cell Carrier-96 black and optically clear bottom plates (PerkinElmer) to reach a density of 15,000 cells per well at the day of infection. HeLa cells were infected as described in the above section.

Post-infection, cells were washed twice with sterile PBS and fixed with 40 µl of freshly prepared 4% paraformaldehyde (Thermo Fisher Scientific) for 10 min. Fixed cells were further washed five times and stored at 4°C in PBS. Fixed cells were first permeabilised with 40 µl of 0.2% Triton X-100 for 3 min, washed thrice with PBS, and incubated 1 hr in 40 µl of blocking solution (PBS-bovine serum albumin [BSA] 3%). Bacteria were detected with polyclonal antibodies raised in rabbits against whole fixed cells of *S. aureus* USA300 strains, JE2::*spa*, BPH2919, and BPH3672 (WEHI antibody technology platform, https://www.wehi.edu.au/research/research-technologies/antibody-technologies). Sera were used at 1:1000, diluted in PBS-BSA 3%, Tween 0.05% for 5 hr at room temperature. Wells were then washed thrice with PBS and incubated 45 min with a secondary antibody (donkey anti-rabbit coupled to Alexa 488, 1:2000 dilution, Invitrogen) in PBS-BSA 3% containing 0.05% Tween-20 (Sigma) and 10% normal donkey serum (Abcam). Wells were washed thrice with PBS and incubated with Phalloidin-TRITC (1:4000) and DAPI (1:4000) (Sigma) in PBS for 15 min. Finally, wells were washed five times with PBS and covered with 200 µl of PBS. Plates were covered with aluminium foil and stored at 4°C until image acquisition on the Operetta microscope.

## Confocal microscopy

HeLa-CCL2 grown on coverslips were infected using the same conditions described above. Coverslips were treated with PBS supplemented with 1% BSA and 0.2% Triton X-100 for 20 min at room temperature to permeabilise cells and incubated overnight in a blocking buffer (PBS supplemented with 1% BSA and 0.1% Tween-20). Coverslips were then probed 1 hr at room temperature with an anti-LAMP1 monoclonal antibody (1:250, clone H4A3 [mouse], Developmental Studies Hybridoma Bank) and 1:1000 polyclonal anti-*S. aureus* diluted in blocking buffer supplemented with 10% normal goat serum (Abcam). Coverslips were washed thrice with PBS then incubated overnight at 4°C with 1:2000 anti-rabbit (488), anti-mouse (647) secondary antibodies diluted in blocking buffer supplemented with 10% normal goat serum. Coverslips were then incubated 7 min with DAPI (1:5000), washed five times and mounted in Prolong Gold antifade (Thermo Fisher Scientific). Samples were imaged on the Zeiss LSM780 confocal microscope.

## High-content imaging acquisition and analysis

Cells were analysed using the Operetta CLS high-content analysis system (PerkinElmer). For each well, images were acquired in a single plane at 11 non-overlapping fields of view (675 × 508 µm/1360 × 1024 pixels in size) using a ×20 PLAN long working distance objective (NA 0.45). DAPI fluorescence (HeLa cell nuclei) was imaged with the filter set: excitation = 360–400 nm, emission: 410–480 nm; 50 ms exposure. A488 fluorescence (*S. aureus*) was imaged with the filter set: excitation = 460–490 nm, emission = 500–550 nm; exposure = 200 ms. A594 fluorescence (HeLa actin stained by Phalloidin-TRITC) was imaged with the filter set 'StdOrange1/Cy3' filter set (excitation: 520–550 nm, emission: 560–630 nm; 0.5-s exposure).

Image processing and analysis were performed using the PhenoLOGIC machine-learning option in the Harmony software (PerkinElmer, v4.1). Nuclei were segmented from the DAPI channel using the 'Find Nuclei algorithm' (Method B, Area filter >40 µm$^2$, Common Threshold of 0.4). Cells were segmented from the A594 channels using the Find Cells algorithm (Method C, Area filter >100 µm$^2$, Common Threshold of 0.5). The A488 signals corresponding to *S. aureus* were further processed using a sliding parabola (curvature, 50 pixels) and Gaussian filter (filter width, 1 pixel) to remove noise and improve the signal-to-noise ratio. *S. aureus* were segmented by applying the Find Spots algorithm (Method A, Relative Spot Intensity 0.280, Splitting Coefficient 0.5).

## Processing of PI fluorescence signals

For every 96-well plate, the PI uptake data for each well at each timepoint were standardised to the JE2 strain control using proportion of maximum scoring (POMS): (PI uptake − min(Pi uptake [JE2]))/range(PI uptake [JE2]). Experiments with less than two JE2 replicates available per plate were excluded from our analyses. Standardised data were used to fit a cubic smoothing spline (*Little, 2013*) using the R function smooth.spline(). Technical replicates within each plate were classified as outliers and excluded if >10% of their timepoint values differed by more than 1.5 times the interquartile range (Tukey method), between the fitted value and the mean for a given isolate. After excluding outlier replicates, fitted data were used to calculate the following PI uptake parameters: AUC, maximum PI uptake rate ($r^{max}$), peak PI uptake ($\mu^{max}$), time to maximum PI uptake rate ($t(r^{max})$), time to peak PI uptake ($t(\mu^{max})$), trough PI uptake, time of trough, and final PI uptake.

## Dimensionality reduction of PI uptake data

Principal component analysis was performed using the 'dudi.pca()' function in the R package 'adegenet' and the randomForest package in R was used for fitting an unsupervised random forest model. We used the similarity matrix generated by the model to define similarity cluster of PI uptake. We used the output of the RF model to calculate the importance of each PI uptake parameter defined as mean decrease in Gini index and mean decrease in accuracy (*Breiman, 2001*).

## PI uptake GWAS

We transformed the mean PI uptake AUC data using the automated normalisation package best-Normalize in R. A GWAS using the normalised AUC data and the 158,169 core-genome variants (all positions where at least 90% of the strains had at least 10 reads coverage, see above) obtained after mapping isolates reads to reference genome BPH2947. All 387 isolates were included in the GWAS, including serial isolates from the same patient, as this outcome is bacterial and not expected to be affected by the host (*Tonkin-Hill et al., 2022*). To correct for the population structure, we used the factored spectrally transformed linear mixed models (FaST-LMM) implemented in *pyseer* v1.3.6 (*Lees et al., 2018*). Random effects in Fast-LMM were computed from a kinship matrix based on the core-genome SNPs generated by Gemma v0.98.1 (*Zhou and Stephens, 2012*). The Bonferroni method was used to correct p values for multiple testing. We performed the GWAS using single variants and the gene-burden test implemented in pyseer. We excluded synonymous mutations for single variants and gene-burden GWAS. For the single variants GWAS, only mutations with a minimum allele fraction (MAF) of 0.01 (as suggested in the pyseer documentation [https://pyseer.readthedocs.io/en/master/index.html]) were kept and at least two independent acquisitions across the phylogeny. For the gene-burden GWAS, we aggregated rare mutations (MAF <0.01), with 3072 coding sequences of the reference and modelled the association of the PI AUC with the presence/absence of any protein-altering mutations in each BPH2947 gene. For consistent annotation of mutations, we identified BPH2947 genes homologs using BLASTP and annotated FPR3757 genes using *aureowiki* (*Fuchs et al., 2018*) and *Microbesonline* (*Alm et al., 2005*). The GWAS analysis was run using a customised in-house pipeline (https://github.com/stefanogg/CLOGEN).

## GWAS power calculation

We used BacGWASim (*Saber and Shapiro, 2020*) to simulate a genotype matrix with 1000 and 150,000 variants for a range of sample size between 300 and 9600. We set mutation rate at 0.06 and recombination rate at 0.01. We simulated 10 phenotype replicates with 16 true causal variants and a heritability of 0.5. We then run pyseer on each replicated simulation using the same pipeline described above. As in *Denamur et al., 2022*, power was calculated as the fraction of identified variants based on a Bonferroni corrected p value below 0.05.

## Determination of genetic pairs with contrasting PI uptake

Isolate pairs for the convergent evolution analysis were identified by screening pairs separated by less than 200 mutations distance for statistically significant differences in the PI uptake AUC (Mann–Whitney test), wherein an isolate causing low PI uptake (isolate-2) was compared to a reference isolate causing higher PI uptake (isolate-1). The genetic distance between closely related isolates was calculated using Snippy and is based on the number of variants identified when mapping the reads of

isolate-2 on the draft assembly of isolate-1 (*Higgs et al., 2022*). To avoid biases related to assembly errors and uneven reads coverage between the two isolates, variant calls were filtered as previously described (https://github.com/stefanogg/staph_adaptation_paper; *Giulieri et al., 2022a*; *Giulieri, 2022*). Non-redundant and phylogenetically independent genetic pairs were identified by manual inspection of the phylogenetic tree.

## Genetic pairs analysis

Mutations identified in genetic pairs and filtered as described above were further characterised using a multilayered annotation strategy as previously described (*Giulieri et al., 2022a*). Firstly, mutated coding regions (amino acid sequences) across draft genomes were clustered using CD-HIT. We then used BLASTP to identify homologs of each cluster within the *S. aureus* USA300 FPR3757 reference genome that was annotated using the AureoWiki repository (*Fuchs et al., 2018*), with a 90% identity and 50% coverage threshold. As genetic pairs were phylogenetically independent and non-redundant, emergence of the same mutation or mutations in the same locus in multiple pairs indicated convergent evolution and was suggestive of positive selection. Based on this, we ranked USA300 FPR3757 homologs according to the number of pairs with mutations.

## Comparison of PI uptake AUC between phylogenetically independent pairs with or without homoplasic mutations

A set of phylogenetically independent pairs was sampled from the tree using the R package ggtree. The impact of the presence/absence of homoplasic mutations on the PI uptake AUC difference within pairs was modelled using linear regression. The sampling and regression were repeated in 100 replicates. As this approach is not suitable for p-value statistical significance, the distribution of the regression coefficient beta of the replicates was used to infer the statistical support for the associations of the homoplasic mutations with the PI uptake phenotype.

## Code and data availability

Scripts to process PI uptake data and to perform genomic analyses are available on github at https://github.com/stefanogg/*InToxSa* (copy archived at *Giulieri and Guerillot, 2023*). The code for genomic analyses is available on https://github.com/stefanogg/CLOGEN (copy archived at *Giulieri, 2023*) (GWAS analysis), and on https://github.com/stefanogg/staph_adaptation_paper (*Giulieri, 2022*) (comparative genomics of genetic pairs).

Whole-genome sequences of the 387 clinical strains are available in the European Nucleotide Archive under Bioproject accession number PRJEB27932.

## Aureusimine B identification

Bacterial extracts were isolated from 30 ml cultures grown in TSB at 37°C overnight under agitation. Bacterial cells were pelleted by centrifugation at 4000 × *g* during 30 min and the culture supernatants were sterilised by passage through a 0.22 mM filter. For each strain, 10 ml of supernatant were added to an equal volume of ethyl acetate in glass tubes, vortexed and allowed to extract at room temperature overnight. Ethyl acetate extracts were dried in vacuo. Dried ethyl acetate extracts were resuspended in 100 ml methanol and 2 ml of each sample was analysed by HPLC using the Shimadzu Prominence HPLC system coupled to a SPD-M20A diode array detector. The column oven (CTO-20A) was set to 40°C and aureusimine B was separated on Kinetex C18, 75 × 3 mm, 2.6 μm column (Phenomenex). Purified aureusimine B was used as reference standard (Bioaustralis Fine Chemicals). All used chemicals were of analytical grade.

Samples were run with water, 0.1% trifluoroacetic acid (solvent A) and acetonitrile (solvent B). The gradient elution was performed on the HPLC at a flow rate of 0.5 ml/min as follows: 10% B for 3.5 min, 10–100% B over 12.5 min, 100–10% B over 1 min, then 10% B for 7 min (total run time, 24 min).

## Additional information

### Funding

| Funder | Grant reference number | Author |
|---|---|---|
| National Health and Medical Research Council | GNT2018880 | Abderrahman Hachani Stefano G Giulieri Romain Guérillot |
| National Health and Medical Research Council | GNT1105525 | Timothy P Stinear |
| National Health and Medical Research Council | GNT1145631 | Timothy P Stinear |
| National Health and Medical Research Council | GNT1196103 | Benjamin P Howden |

The funders had no role in study design, data collection, and interpretation, or the decision to submit the work for publication.

### Author contributions

Abderrahman Hachani, Conceptualization, Resources, Data curation, Formal analysis, Supervision, Funding acquisition, Validation, Investigation, Visualization, Methodology, Writing – original draft, Project administration, Writing – review and editing; Stefano G Giulieri, Conceptualization, Resources, Data curation, Formal analysis, Validation, Investigation, Methodology, Writing – original draft, Writing – review and editing; Romain Guérillot, Conceptualization, Resources, Data curation, Formal analysis, Validation, Investigation, Visualization, Methodology, Writing – review and editing, Supervision; Calum J Walsh, Ye Mon Soe, Ashleigh S Hayes, Investigation; Marion Herisse, Sacha Pidot, Investigation, Methodology, Writing – original draft; Sarah L Baines, Resources, Data curation, Writing – original draft; David R Thomas, Formal analysis, Investigation; Shane Doris Cheung, Data curation, Methodology; Ellie Cho, Investigation, Methodology; Hayley J Newton, Investigation, Writing – original draft; Ruth C Massey, Resources, Data curation; Benjamin P Howden, Resources, Supervision, Funding acquisition, Writing – review and editing, Joint senior author; Timothy P Stinear, Resources, Supervision, Funding acquisition, Visualization, Writing – original draft, Project administration, Writing – review and editing

### Author ORCIDs

Abderrahman Hachani http://orcid.org/0000-0001-8032-2154
Stefano G Giulieri http://orcid.org/0000-0001-5366-1943
Romain Guérillot http://orcid.org/0000-0001-9915-1420
Ye Mon Soe http://orcid.org/0000-0003-4776-2523
Sarah L Baines http://orcid.org/0000-0002-0557-0518
Shane Doris Cheung http://orcid.org/0000-0001-8733-4756
Ashleigh S Hayes http://orcid.org/0000-0003-2926-7561
Ellie Cho http://orcid.org/0000-0002-2932-1890
Hayley J Newton http://orcid.org/0000-0002-9240-2001
Benjamin P Howden http://orcid.org/0000-0003-0237-1473
Timothy P Stinear http://orcid.org/0000-0003-0150-123X

### Decision letter and Author response

Decision letter https://doi.org/10.7554/eLife.84778.sa1
Author response https://doi.org/10.7554/eLife.84778.sa2

## Additional files

### Supplementary files

• Supplementary file 1. Parameters data: This is the intracellular toxicity of *S. aureus* (*InToxSa*) data used to create the propidium iodide (PI) uptake plots in *Figure 1C, D*. Z' factor: These are the Z' factors calculated from the parameters data. Legend: Variable details.

• Supplementary file 2. Table of count data used for *Figure 2B*.

• Supplementary file 3. Dataset: Basic table describing the 387 isolates characterised in this study. Legend: Variable details.

• Supplementary file 4. Dataset: Basic table of the propidium iodide (PI) uptake values for the 387 isolates. Legend: Variable details.

• Supplementary file 5. The genome-wide association study (GWAS) data transformed propidium iodide (PI) uptake data, multidimensional scaling (MDS) lineage data, lineage effect tests, gene-burden test data, and variable legend.

• Supplementary file 6. Convergence analysis data.

• Supplementary file 7. List of primers used to introduce convergent mutation by site-directed mutagenesis in BPH3370.

• Supplementary file 8. Output of the comparison of the comparison of absolute propidium iodide (PI) uptake area under the curve (AUC) difference between pairs with or without mutations in the most convergent genes identified from the convergence evolution analyses. This analysis was performed in 100 replicated sets of phylogenetically independent genetic pairs.

• MDAR checklist

### Data availability

-The genome sequences of *Staphylococcus aureus* clinical isolates are deposited on the European Nucleotide Archive under Bioproject accession number PRJEB27932.- All data generated during this study are included in the supplementary files of the manuscript. The generated codes are publicly available on: https://github.com/stefanogg/InToxSa (copy archived at *Giulieri and Guerillot, 2023*).

The following previously published dataset was used:

| Author(s) | Year | Dataset title | Dataset URL | Database and Identifier |
|---|---|---|---|---|
| Giulieri SG | 2018 | Genomic Exploration of Sequential Clinical Isolates Reveals a Distinctive Molecular Signature of Persistent *Staphylococcus aureus* Bacteraemia | https://www.ebi.ac.uk/ena/browser/view/PRJEB27932 | European Nucleotide archive, PRJEB27932 |

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
