## [Editor Report]

This paper describes a new method to investigate *Staphylococcus aureus* intracellular virulence that has produced important insights into the mechanisms of staphylococcal pathogenesis. The results are convincing and the methodology is state-of-the-art. The authors have responded to the reviewer comments and resolved the issues identified during the review. This paper will be of interest to scientists studying microbial intracellular pathogenesis and cell biology.

---

## [Decision Letter]

**Decision letter after peer review:**

Thank you for submitting your article "A high-throughput cytotoxicity screening platform reveals agr-independent mutations in bacteraemia-associated *Staphylococcus aureus* that promote intracellular persistence" for consideration by *eLife*. Your article has been reviewed by 2 peer reviewers, and the evaluation has been overseen by a Reviewing Editor and Arturo Casadevall as the Senior Editor. The following individual involved in the review of your submission has agreed to reveal their identity: Matthew Culyba (Reviewer #2).

Essential revisions:

*Reviewer #1 (Recommendations for the authors):*

Consider depositing the long read sequence of the HeLa strain, or at least chromosome mapping/ structure variant mapping. These could be important when other investigators attempt to reproduce results using their version of the cell line.

Line 145 – What specifically is "very low intra assay variation"?

Line 283 – Unclear if these are the only 28 pairs that met the criterion, or if there were others but that weren't selected.

Line 290 – Were homoplasic mutations in these loci found between other pairs of strains (ie where there were no significant differences)

Line 324-328 – This correlation between π AUC and intracellular bacterial titer should be shown as a scatterplot for clarity

*Reviewer #2 (Recommendations for the authors):*

1. The InToxSa assay was devised to reflect the toxicity of intracellular *S. aureus*, however, it is not made clear in the manuscript that the intracellular nature of this assay is actually an important feature that distinguishes it from what would be observed by testing the same strains in the equivalent extracellular toxicity assay. For example, the interpretation of the InToxSa versus Tryptan blue exclusion assay data was that InToxSa was more sensitive. However, two key variables differed in this comparison: (1) extracellular toxicity vs intracellular toxicity and (2) cell type used. Is InToxSa more sensitive in this comparison because of the intracellular nature of the assay, or because it employs HeLa cells instead of THP1 macrophages? Maybe HeLa cells are just more sensitive regardless? What would happen if you repeated the Tryptan blue exclusion assay with HeLa cells?

2. High throughput screening assays often report a 'reconfirmation rate' for the assay. This helps address day-to-day assay signal variation from the actual experimental samples (instead of controls). If you just rerun the 56 isolates that make up the 28 pairs from the convergence analysis, how well do the data replicate in terms of the difference in π uptake? I'm wondering how much of the low validation/confirmation rate with the NTLM and allelic exchange strains could be due to day-to-day assay noise in the primary screen. Maybe some of these 'hits' were simply false positives?

3. The analysis focuses on negative π difference values. What is the interpretation of a positive value for π differences? Higher toxicity? Were any of these identified by GWAS or the convergence analysis? Are these of biological interest?

4. I'm wondering how the authors envision others deploying this system to screen clinical isolates since the 'gene burden GWAS' only found agr, which is a well-described virulence locus. How many isolates would have to be screened and analyzed in this manner to find additional biologically relevant mutant alleles? The statistical power of this approach is related to the number of isolates (N), their genetic relatedness, the magnitude of the effect size (in this case, π uptake signal), and the gene burden. A power analysis could be a useful way how to think about deploying this assay using the 'phenomics' approach. Would it be possible to model these parameters in the gene burden GWAS, and provide an estimate of the number of isolates (N) that would need to be screened using the assay to find mutant alleles as a function of their effect size (i.e. PI-uptake), gene burden, and the genetic relatedness of the sample population? Perhaps you will find that the assay is best deployed in settings where the strains being screened are all very closely genetically related (e.g. serial isolates from the same patient or transmission outbreak). Outside of these situations, I wonder if N might be too large for this to be a feasible approach to finding novel genes involved in intracellular toxicity. Modeling this as a power analysis would describe this quantitatively and likely point to the best types of clinical strain samples to pursue given the practical constraint that screening more than a few thousand isolates in this system would not be feasible. It seems you already took steps to enrich for serial isolates from the same patient and then the convergence analysis was even further enriched for these samples. These are clearly critical steps to get good information output. Modeling the statistical power of this would be a nice complement I think.

5. All of the 'hits' from the gene burden GWAS and convergence analyses were attempted to be validated using the corresponding strains from the NTML. This begs the question: why not just screen the entire NTML as a starting point? It seems this approach would have not only offered better validation for the utility of the assay but also provided maximal statistical power for genotype-phenotype correlations given the isogenic background within this strain library. With this in mind, can you elaborate further on your choice of approach?

6. The manuscript describes using HeLa cells as an in vitro model system for professional phagocytic cells that are thought to be important for *S. aureus* clearance in vivo. The biological relevance of the HeLa cell system would be further supported by also studying some mutants in a system that employs professional phagocytes to study intracellular *S. aureus* persistence. For example, it would be of interest to know if the ausA mutants have a persistence phenotype in a macrophage or not.

7. The statistical 'gene burden GWAS' procedure only identified agr as having a significant P-value. In contrast, a manual 'convergence' analysis and hunt for homoplastic mutations for π uptake seemed to identify additional plausible candidates, but this manual procedure was not supported by statistical analysis. Can you explain why the convergence analysis would find loci that the 'gene burden GWAS' failed to identify? Can the mutations identified by the convergence analysis be further supported within a formal statistical framework? In the 'convergence' analysis, you only consider pairs with <200 SNP differences, which effectively increases the chance (i.e. statistical power) that any given mutation could be responsible for the observed π uptake effect size. What would happen if you re-ran the 'gene burden GWAS' using only a subset of the more genetically related strain pairs? Would that increase sensitivity enough to identify the homoplastic mutations you found in the 'convergence' analysis?

---

## [Author Response]

Essential revisions:Reviewer #1 (Recommendations for the authors):Consider depositing the long read sequence of the HeLa strain, or at least chromosome mapping/ structure variant mapping. These could be important when other investigators attempt to reproduce results using their version of the cell line.

We refer interested investigators to the American Type Culture Collection (ATCC) from where the HeLa CCL2 were obtained and refer them to the following manuscript describing the phenotypes obtained during bacterial infection of various HeLa cell types (Tang L. et al., DOI: 10.1038/s41592-019-0375-1). The haplotype-resolved genome and epigenome of the aneuploid HeLa CCL2 cell line used in this study can be found in the following publication: DOI: 10.1038/nature12064.

Line 145 – What specifically is "very low intra assay variation"?

We substantiate “very low intra assay variation” by providing the coefficient of variation (CoV) and standard deviation (Std. Dev.) values in Table 1. The CoV values are lower than those recommended in the ISO standards for diagnostic tests. We now also provide standard deviation values (Std. Dev.) in Table 1 as requested.

Line 283 – Unclear if these are the only 28 pairs that met the criterion, or if there were others but that weren't selected.

Pairs with contrasting π uptake were selected according to 3 criteria:

1) < 200 mutations distance between isolate 1 and isolate 2,

2) significantly lower π uptake in isolate 2 compared to isolate 1 (Mann-Whitney test), and

3) phylogenetic independence (i.e pairs belonging to non-overlapping clades) (Methods lines 691-701).

While the total number of pairs satisfying criteria 1 and 2 is high, the need for phylogenetic independence dramatically restricts the number of eligible pairs. Without this independence filter, the occurrence of homoplasic mutations would be unduly inflated according to the size of the clade and meaningless for detecting and comparing the strength of the genetic association signals (i.e representing redundant genetic comparisons). Previous studies have successfully used the same approach (Pidot et al., 2018, DOI: 10.1126/scitranslmed.aar6115) to address this issue.

Line 290 – Were homoplasic mutations in these loci found between other pairs of strains (ie where there were no significant differences)

This is an interesting question that we have now addressed by comparing the difference in π uptake between independent genetic pairs that carried mutations in the most convergent genes and those that didn’t. Because multiple combinations of phylogenetically independent pairs are possible, we repeated the analysis in 100 replicates and calculated the median of absolute difference in π uptake for each group and fitted 100 replicated linear regressions to assess the strength of the association between the presence/absence of homoplasic mutations and the π uptake difference.

For six genes (agrA, glcA, ribA, fmtB, sbnF, tarL’), this supplementary analysis provided full support of the association with the cytotoxicity phenotype (100% of the linear regression replicates had a coefficient above 0). Similarly, 84% replicates supported the association between mutations occurring in the gene ausA and the π uptake. This new analysis is reported in the results (lines 319-339), methods (lines 780-787) and illustrated in Figure 5 —figure supplementary 2.

Together, our analyses show that most genes with homoplasic mutations in phenotypically divergent pairs were not mutated in phenotypically similar pairs, with the gene agrC being the only exception. While the agrC transposon mutant has reduced cytotoxicity, confirming the expected role of agrC loss-of-function mutations in reduced cytotoxicity, it remains possible that the mutations identified in our clinical strains could be compensatory and therefore not directly associated with the phenotype (see the following references reporting such phenomena, DOI: 10.1128/IAI.003331-18 and DOI: 10.7554/*eLife*.77195) or, that the null effect on cytotoxicity of aggregated agrC mutations could be due to the presence of both cytotoxicity-increasing and decreasing mutations within the same isolate. We have now included these supporting references in the text (Lines 444-447).

Line 324-328 – This correlation between π AUC and intracellular bacterial titer should be shown as a scatterplot for clarity

We now provide two scatterplots showing the statistically significant inverse Pearson correlations of: (1) π AUC with the percentage of infected cells; and (2) π AUC with the number of *S. aureus* per infected cell at 24 hours post infection. These two plots are cited in the main text and available in Figure 5 –Supplementary figure 5.

Reviewer #2 (Recommendations for the authors):1. The InToxSa assay was devised to reflect the toxicity of intracellular *S. aureus*, however, it is not made clear in the manuscript that the intracellular nature of this assay is actually an important feature that distinguishes it from what would be observed by testing the same strains in the equivalent extracellular toxicity assay. For example, the interpretation of the InToxSa versus Tryptan blue exclusion assay data was that InToxSa was more sensitive. However, two key variables differed in this comparison: (1) extracellular toxicity vs intracellular toxicity and (2) cell type used. Is InToxSa more sensitive in this comparison because of the intracellular nature of the assay, or because it employs HeLa cells instead of THP1 macrophages? Maybe HeLa cells are just more sensitive regardless? What would happen if you repeated the Tryptan blue exclusion assay with HeLa cells?

We are aware of the potential limitations of both Trypan blue exclusion and InToxSa methods, and these have been presented in the introduction (Lines 65-97) and in the discussion of the manuscript (Lines 403-428). In Figure 3, the InToxSa assay is more sensitive than the trypan blue exclusion assay in the sense that InToxSa detects more differences between isolates and thus increases the power of our analytical framework to identify genetic differences between these isolates. It would be possible to measure the sensitivity of HeLa cells to extracellular, secreted staphylococcal toxins using Trypan Blue, but such experiments are outside the aims of the current study. Furthermore, such assays are low-throughput in nature and are not continuous (end-point only), requiring extensive trial-and-error experimentation to establish consistent timepoints that represent thresholds of cytotoxicity between isolates.

2. High throughput screening assays often report a 'reconfirmation rate' for the assay. This helps address day-to-day assay signal variation from the actual experimental samples (instead of controls). If you just rerun the 56 isolates that make up the 28 pairs from the convergence analysis, how well do the data replicate in terms of the difference in π uptake? I'm wondering how much of the low validation/confirmation rate with the NTLM and allelic exchange strains could be due to day-to-day assay noise in the primary screen. Maybe some of these 'hits' were simply false positives?

The 56 isolates were assessed on at least two different plates on different days, with three biological replicates on each plate (2 to 5 plates/days per isolate; 6 to 15 replicates in total). Day-to-day assay noise is not responsible for the low reproduction of phenotype in mutants. We believe that this low confirmation rate is most likely due to epistatic interactions where the functional impact of the detected genomic signatures is conditioned upon the presence of other interacting gene(s)/mutation(s) in a specific genetic background. Unfortunately, the Nebraska transposon mutant library has been made in a ST8 background but none of the convergent mutations identified in our study have emerged in this genetic background.

3. The analysis focuses on negative π difference values. What is the interpretation of a positive value for π differences? Higher toxicity? Were any of these identified by GWAS or the convergence analysis? Are these of biological interest?

We focussed on pairs with loss of cytotoxicity to be consistent with the hypothesis that loss of toxicity (rather than gain) is an adaptive phenotype during infection. This is supported by our gene-burden GWAS, where 10/10 most significant hits are associated with loss of cytotoxicity (mean normalised π AUC decrease = –0.8, Supplementary file 5). However, we agree that there could also be an interest in investigating gain of cytotoxicity, as this phenotype might be advantageous for the bacteria in certain clinical settings (e.g. abscess formation). To further explore this, we selected 7 genetic pairs that had a significant π uptake increase from iso1 to iso2 and were phylogenetically independent from each other and from the previously selected pairs. We repeated the convergence analysis and found that adding these pairs further increased the number of independent (homoplasic) ausA and fmtB mutations. This reinforces the importance of ausA as an agr-independent cytotoxicity locus (Result: lines 320-329; Figure 5- Supplementary Figure 5).

4. I'm wondering how the authors envision others deploying this system to screen clinical isolates since the 'gene burden GWAS' only found agr, which is a well-described virulence locus. How many isolates would have to be screened and analyzed in this manner to find additional biologically relevant mutant alleles? The statistical power of this approach is related to the number of isolates (N), their genetic relatedness, the magnitude of the effect size (in this case, π uptake signal), and the gene burden. A power analysis could be a useful way how to think about deploying this assay using the 'phenomics' approach.

To address this issue, we have now used the bacterial GWAS power calculation approach suggested in Denamur et al., PLoS Genetics 2022 (this reference is included in the revised manuscript and in the methods) and used the simulator BacGWASim to simulate genome datasets of increasing size (from 300 samples to 9600) for a genome length of 1 million bp, a mutation rate of 0.06 and recombination rate of 0.01. For each dataset 16 causal variants were distributed in 10 genomes. We then ran pyseer using the same approach as in our GWAS analysis and calculated power as the proportion of causal variants that were above the Bonferroni-corrected threshold. We found that the power of the mutations GWAS was overall low, with 0% mutations called for sample size < 2,400 and a 20% power for the largest sample. As commented by the reviewer and consistent with the BacGWASim paper (Saber, Morteza and Shapiro, Microb Genom 2020), bacterial GWAS has limited power when applied to low frequency recombining pathogen like *S. aureus*. This supports our strategy to use convergent evolution analysis in our exploration of mutations associated with cytotoxicity. Despite not being specifically designed for convergence evolution analysis, our simulations for a low variants GWAS (n=1,000) further confirms our strategy (see Figure 4 – Supplementary Figure 4).

Note however, that this power calculation approach has inherent limitations. Firstly, the magnitude of the functional impact of mutations is not accounted for. Secondly, it relies heavily on rough estimates of core genome sizes, recombination and mutation rates that would be dataset dependent. Finally, it doesn’t simulate rare variants and the impact of aggregating them in a gene burden test. We agree with the reviewer that a more sophisticated bacterial GWAS simulation framework would provide more accurate information to plan bacterial phenotype-genotype studies. However, we feel that a more detailed power analysis is outside the aims of the current study and thus should be the topic of a specific statistical genomics methods paper.

5. Would it be possible to model these parameters in the gene burden GWAS, and provide an estimate of the number of isolates (N) that would need to be screened using the assay to find mutant alleles as a function of their effect size (i.e. PI-uptake), gene burden, and the genetic relatedness of the sample population?

We agree with the reviewer that modelling these parameters in the gene burden GWAS would be a useful and complementary approach to model the statistical power of such studies. However, such analysis is outside the aims of the current study and would represent a standalone statistical genomics methods paper.

6. All of the 'hits' from the gene burden GWAS and convergence analyses were attempted to be validated using the corresponding strains from the NTML. This begs the question: why not just screen the entire NTML as a starting point? It seems this approach would have not only offered better validation for the utility of the assay but also provided maximal statistical power for genotype-phenotype correlations given the isogenic background within this strain library. With this in mind, can you elaborate further on your choice of approach?

We acknowledge the reviewer’s point. A previous study has successfully identified mutations associated with intracellular persistence using libraries of transposon mutants (https://doi.org/10.1073/pnas.1520255113). However, the focus of our study was to identify naturally occurring and potentially clinically important adaptive mutations directly from a cohort of clinical isolates.

7. The manuscript describes using HeLa cells as an in vitro model system for professional phagocytic cells that are thought to be important for *S. aureus* clearance in vivo. The biological relevance of the HeLa cell system would be further supported by also studying some mutants in a system that employs professional phagocytes to study intracellular *S. aureus* persistence. For example, it would be of interest to know if the ausA mutants have a persistence phenotype in a macrophage or not.

The manuscript does not describe HeLa cells as an in vitro model for professional phagocytes. We agree with the reviewer regarding the verification of the phenotypic outputs identified using HeLa cells in professional phagocytes, which is the focus of future work.

8. The statistical 'gene burden GWAS' procedure only identified agr as having a significant P-value. In contrast, a manual 'convergence' analysis and hunt for homoplastic mutations for π uptake seemed to identify additional plausible candidates, but this manual procedure was not supported by statistical analysis. Can you explain why the convergence analysis would find loci that the 'gene burden GWAS' failed to identify? Can the mutations identified by the convergence analysis be further supported within a formal statistical framework?

We believe that our convergence analysis provides a complementary approach to allele-counting GWAS methods like that implemented in pyseer. The main advantage of convergence-based methods is the elimination of biases due to the underlying population structure. Pyseer uses a linear mixed model to correct for population structure, whereby the genetic relatedness of the isolates is expressed as kinship matrix and modelled as random effect. This method is widely used in human GWAS and might be sufficient for highly recombinogenic bacteria such as Streptococcus pneumoniae (see for example Lees et al., Nat Commun 2019). However, it is possible that this indirect correction for population structure is not complete in *S. aureus*, preventing the discovery of new genetic loci when performing a GWAS. By contrast, convergent evolution-based analysis eliminates biases due to the populations structure, with lower risk of identifying false positive. This is demonstrated by a large number of studies performed by both our group (Howden, PLoS Pathogen 2011; Pidot, Sci Transl Med 2018) and others (Das, PNAS 2016), who were able to identify biologically meaningful mutations from genetically close but-phenotypically divergent isolates. These studies were able to identify new phenotype-genotype associations without a statistical framework. However, we agree with the reviewer that providing statistical support would improve the confidence in the strength of the association and thus help prioritising genes for further investigation. In our response to Reviewer 1, we have now provided solid support for the contributions of 7 out of 20 genes (including agrA and ausA) in affecting intracellular cytotoxicity (see Figure 5 and Figure 5 Supplementary figure 1).

9. In the 'convergence' analysis, you only consider pairs with <200 SNP differences, which effectively increases the chance (i.e. statistical power) that any given mutation could be responsible for the observed π uptake effect size. What would happen if you re-ran the 'gene burden GWAS' using only a subset of the more genetically related strain pairs? Would that increase sensitivity enough to identify the homoplastic mutations you found in the 'convergence' analysis?

Whilst we agree with the importance of providing statistical support for our convergent evolution analysis, we feel that using an allele-counting approach like the one implemented in pyseer or other similar packages wouldn’t be appropriate for a small dataset of 56 paired closely related strains. Firstly, such approach would lead to drastic loss of power given the inclusion of 7-times less strains that in the original GWAS. Secondly the correction for population structure implemented in pyseer is not designed for pairs of closely related isolates. As mentioned above, we now provide a robust framework to support the associations between homoplasic mutations and cytotoxicity changes.